# NMA-tune: Generating Highly Designable and Dynamics Aware Protein Backbones

Urszula Julia Komorowska [1]   Francisco Vargas [1]   Alessandro Rondina [2]   Pietro Lio [1]   Mateja Jamnik [1]

## Abstract

Protein's backbone flexibility is a crucial property that heavily influences its functionality. Recent work in the field of protein diffusion probabilistic modelling has leveraged Normal Mode Analysis (NMA) and, for the first time, introduced information about large scale protein motion into the generative process. However, obtaining molecules with both the desired dynamics and designable quality has proven challenging. In this work, we present NMA-tune, a new method that introduces the dynamics information to the protein design stage. NMA-tune uses a trainable component to condition the backbone generation on the lowest normal mode of oscillation. We implement NMA-tune as a plug-and-play extension to RFdiffusion, show that the proportion of samples with high quality structure and the desired dynamics is improved as compared to other methods without the trainable component, and we show the presence of the targeted modes in the Molecular Dynamics simulations.

## 1. Introduction

Generative AI has had a tremendous impact on the field of protein design in the recent years. Denoising diffusion probabilistic models (DDPMs), such as RFdiffusion (Watson et al., 2023), Chroma (Ingraham et al., 2023) or FrameDiff (Yim et al., 2023b), have been applied to design novel protein backbones with given *structural* properties. A common task is to perform *motif scaffolding*, that is, to design a stable protein backbone that consists of a pre-defined motif and a scaffold that holds the motif in place.

Motif scaffolding encompasses various other tasks ranging from designing novel antibodies (Correia et al., 2014) to scaffolding enzyme active sites (Watson et al., 2023).

However, often the biological mechanisms behind protein functionality depend not only on the structure, but also on the dynamics (Berendsen & Hayward, 2000). Currently, the beginning stage of the design, more than validation and refinements stages, is limited by the available tools to include dynamics constraints. Often Molecular Dynamics (MD) simulations are ran to understand the relevant motions, then one makes an informed change in the current design model and runs the MD simulation again (Childers & Daggett, 2017). Since it is not immediately known how the MD trajectory will change under a specific structure modification, this step must be reiterated many times, which is inefficient and costly.

Recently, a new method emerged that overcomes this design→assess→refine loop. Komorowska et al. (2024) introduce *dynamics conditioning*, a way to design a protein backbone whose chosen subset of $C_\alpha$ carbons have user-specified displacements in the lowest non-trivial normal mode, which corresponds to the large scale collective motion. They capture the coarse-grained dynamics using the Normal Mode Analysis (NMA) (Bahar & Rader, 2005), a computational method to obtain the eigenvectors of motion under an assumed force-field. This is extended to *joint conditioning*, which is structure and dynamics conditioning applied together. Here, for ease, we call the dynamics conditioning from Komorowska et al. (2024) *NMA-guidance*.

NMA-guidance takes first steps towards dynamics-informed design, but also suffers from a number of shortcomings. In DDPMs, the protein generation is driven by the unconditional term (does the new sample resemble real protein?) and conditional term (does the sample meet the imposed conditions?). Previous work uses a simple analytical function to link the dynamics of the protein with the probability of sampling such protein under conditional real data distribution. This method, while beneficial from the perspective of the model design and generalisability, is sensitive to the user defined sampling parameters. Moreover, the generation of high quality samples with both desired structural and dynamical properties has not yet been achieved. This

---

[1]Department of Computer Science and Technology, University of Cambridge, Cambridge, United Kingdom [2]Department of Molecular and Translational Medicine, University of Brescia, Brescia, Italy. Correspondence to: Urszula Julia Komorowska <ujk21@cam.ac.uk>.

*Proceedings of the 42nd International Conference on Machine Learning*, Vancouver, Canada. PMLR 267, 2025. Copyright 2025 by the author(s).

could be attributed to the significant challenge in structure conditioning with motifs as implemented in Komorowska et al. (2024).

To address the difficulty of balancing conditional and unconditional term (as in NMA-guidance), we develop a new method for dynamics conditioning that creates designable scaffolds with given dynamical properties. Our approach retains the advantages of analytical approximation to the conditional probability and the usage of some pre-trained unconditional model, but crucially, we add a novel trainable component that significantly corrects the conditional term. We call this new methodology *NMA-tune*. In contrast to the previous work on the dynamics conditioning that relied on the weighted sum of the unconditional noise and the educated guess of the conditional noise, NMA-tune learns their best combination. The conditioner is a small neural network (530K parameters in our experiments), requires little computational resources to train, allows for faster computation of the loss-guidance term and therefore decreases sampling time, and lastly offers significant improvement in joint conditioning efficiency.

Our contributions are as follows:

- We introduce NMA-tune, a method for turning an unconditional diffusion model into dynamics-conditional one.

- We provide the conditioner as a ready to use plug-in to the open-source model RFdiffusion. Since RFdiffusion is already able to do motif scaffolding, this constitutes an extension to joint conditioning that can easily be used by the wider research community.

- We evaluate the effectiveness of the conditioner and its impact on the sample designability using 3 proteins chosen from the literature. We found that for the dynamics conditioned samples, there exist protein sequences that will fold to protein backbones with desired normal modes, and NMA-tune outperforms existing state-of-the-art.

- We run MD simulations on selected samples and perform Principal Component Analysis (PCA) on their trajectories. We observed higher overlap between the targeted normal mode and largest principal component for the dynamics conditioned samples. This orthogonal validation of NMA trajectories indicates our method translates to success in downstream, biological tasks.

## 2. Background and related work

Since we utilise NMA and diffusion based and score models in our methodology, we start by providing an overview of their underlying theory, and then discuss the specifics of the RFdiffusion model.

### 2.1. Normal Mode Analysis

Normal Mode Analysis (Bahar & Rader, 2005; Bahar et al., 1997; 2010) is a computationally efficient technique to describe functionally relevant protein motions without running more expensive Molecular Dynamics simulation. Protein residues are often represented as a coarse-grained system of $N$ $C_\alpha$ atoms, with positions given by flattened vector of coordinates $x \in \mathbb{R}^{3N}$, and strengths of interactions given by the matrix $\mathbf{K} \in \mathbb{R}^{3N \times 3N}$. NMA assumes the atoms reside near the energy minima and undergo harmonic motions about them. The equation of the harmonic motion for the entire system can be written as $\mathbf{M}\ddot{x} = -\mathbf{K}x$, where $\mathbf{M} \in \mathbb{R}^{3N \times 3N}$ is a mass matrix. Eigenvectors solving this equation describe the amplitudes and frequencies of oscillations, as well as the directions of displacements of the individual atoms. NMA has been empirically verified to correctly describe a wide range of motions in proteins (Bauer et al., 2019). Importantly, often the lowest non-trivial modes of oscillations capture the large-scale collective motions, and a subset of the lowest modes can be sufficient to explain proteins' dynamics (Bauer et al., 2019; Tama & Sanejouand, 2001).

### 2.2. Diffusion probabilistic models

DDPMs (Ho et al., 2020; Sohl-Dickstein et al., 2015) are a class of diffusion generative models able to transform a sample from a standard normal distribution $x_T \sim \mathcal{N}(0, 1)$ into a sample from ground truth data distribution $p_0$. The key observation is that one can transform $x_0 \sim p_0$ into $x_T \sim \mathcal{N}(0, 1)$ by gradually adding Gaussian noise, and then a neural network can learn to reverse this process. The noise is firstly added according to the variance schedule $\{\beta_t\}$ that specifies the noise magnitudes for each step $t \in [0, T]$. For each noising step $p_t(x_t|x_{t-1}) = \mathcal{N}(x_t, \sqrt{1 - \beta_t}x_{t-1}, \beta_t I)$, and the marginal probability is $p_t(x_t|x_0) = \mathcal{N}(x_t, \sqrt{\bar{\alpha}_t}x_0, (1 - \bar{\alpha}_t)I)$, where $\bar{\alpha}_t = \prod_i^t \alpha_i$ and $\alpha_i = 1 - \beta_i$. The noisy sample has a neat form of $x_t = \sqrt{\bar{\alpha}_t}x_0 + \sqrt{1 - \bar{\alpha}_t}\epsilon$, where $\epsilon \sim \mathcal{N}(0, 1)$. It was shown that the above noising can be reversed (Sohl-Dickstein et al., 2015), and the reverse transition densities are given by $p(x_{t-1}|x_t, x_0) = \mathcal{N}(\frac{1}{\sqrt{\bar{\alpha}_t}}(x_t - \frac{1-\alpha_t}{\sqrt{1-\bar{\alpha}_t}}\epsilon), \frac{1-\bar{\alpha}_{t-1}}{1-\bar{\alpha}_t}\beta_t I)$. However, the above forward and reverse densities are tractable only if conditioned on $x_0$, while by definition, there is no information about $x_0$ when sampling novel data. This problem can be circumvented with the appropriate denoiser training objective. The denoiser $\epsilon_\theta$ can be trained with the noise-matching loss (Ho et al., 2020) $L = \mathbb{E}_{t,x_0,\epsilon} ||\epsilon - \epsilon_\theta(\sqrt{\bar{\alpha}_t}x_0 + \sqrt{1 - \bar{\alpha}_t}\epsilon)||^2$. $\epsilon_\theta$ predicts the added noise $\epsilon$ given a noisy version of $x_0$. With this formulation, the denoiser indirectly learns the conditional $p_t(x_0|x_t)$. In the generation process, it iteratively makes guesses about $x_0$ given $x_t$, starting from the fully noised $x_T$.

Score models (Hyvärinen, 2005; Vincent, 2011; Song & Ermon, 2019; Song et al., 2021) operate on similar principles to transform a sample from $p_0$ into noisy $x_T$. In a diffusion based score model (Song et al., 2021), both noising and denoising are expressed as the continuous-time Stochastic Differential Equations (SDEs), with forward $dx = -\frac{1}{2}\beta(t)x dt + \sqrt{\beta(t)}dw$ and the reverse $dx = \left[-\frac{1}{2}\beta(t)x - \beta(t)\nabla_x \ln p_t(x)\right]dt + \sqrt{\beta(t)}d\bar{w}$, where $dw$ and $d\bar{w}$ are the Wiener process in forward and reverse time. Score models and DDPMs can be unified into a single framework using the score-noise equivalence $\nabla_{x_t} \ln p_t(x_t|x_0) = -\epsilon_t/\sqrt{1-\bar{\alpha}_t}$ (for proof, see for example, Appendix F in (Komorowska et al., 2024)), such that any model that is a score predictor is a noise predictor as well.

## 2.3. Loss-guided diffusion

Often, the task is not simply to take a sample from $p_0$, but to sample from $p_0$ subject to conditions $y$. The common methods for the conditional sampling include classifier guidance (Dhariwal & Nichol, 2021), classifier-free guidance (Ho & Salimans, 2022) and loss-guidance (Song et al., 2023). In those frameworks, the reverse SDE requires conditional $\nabla_{x_t} \ln p_t(x_t|y)$, where $y$ is the target quantity, and $p_t(x_t|y)$ is the probability density of $x_t$ given that $x_0$ is subject to $y$. Note that by Bayes' rule $\nabla_{x_t} \ln p_t(x_t|y) = \nabla_{x_t} \ln p(y|x_t) + \nabla_{x_t} \ln p_t(x_t)$.

Both classifier guidance and classifier-free guidance rely on the assumption that the denoiser is able to learn the connection between $x_t$ and $y$. When this connection is difficult to learn, such as in the case when $y$ is the eigenvector of Hessian matrix arising from inter-molecular forces, loss-guidance is more suitable. The first step in the loss-guided diffusion is to specify the model $p(x_0|y) = \frac{p(x_0)\exp[-l_y(x_0)]}{Z_y}$ and $Z_y = \int p(x_0)\exp[-l_y(x_0)]dx_0$, where $l_y(x_0)$ is a custom loss function enforcing the condition $y$ at $t = 0$, and $Z_y$ is the normalisation constant. This additional loss term turns the unconditional sampling distribution into a conditional one. $p(y|x_0)$ is key in finding the loss-guidance term $\nabla_{x_t} \ln p(y|x_t)$. Consider the integral

$$p(y|x_t) = \int_{x_0} p(y|x_0)p(x_0|x_t)dx_0. \tag{1}$$

Although intractable, it can be approximated using a point estimate of the mean of the posterior $p(x_0|x_t)$, as done by Chung et al. (Chung et al., 2022)

$$\hat{x}_0 := \mathbb{E}[x_0|x_t] = \frac{1}{\sqrt{\bar{\alpha}_t}}(x_t + (1-\bar{\alpha}_t)\nabla_{x_t}\ln p(x_t)), \tag{2}$$

$$p(y|x_t) \approx \int_{x_0} p(y|x_0)[\delta(\hat{x}_0) - x_0]dx_0, \tag{3}$$

hence

$$\nabla_{x_t} \ln p(y|x_t) \approx \nabla_{x_t} \ln p(y|\hat{x}_0) =$$
$$= \nabla_{x_t} \ln \exp[-l_y(\hat{x}_0)] = -\nabla_{x_t}l_y(\hat{x}_0). \tag{4}$$

This formulation of loss-guidance is crucial for our method of dynamics conditioning, where we guide the sampling with the custom loss function.

## 2.4. Related work and RFdiffusion

**Protein DDPMs** such as FrameDiff (Yim et al., 2023b), Genie (Lin & AlQuraishi, 2023), RFdiffusion (Watson et al., 2023) or Chroma (Ingraham et al., 2023) were found to produce designable protein backbones both in the unconditional generation and in generation constrained with properties such as symmetry or presence of a given substructure. Didi et al. (2024) offer an extensive survey of structure conditioning methods, and additionally propose to use Doob's h-transform in the structure-conditional training (motif amortisation). Yim et al. (2023a) use Continuous Normalising Flow (Chen et al., 2018) trained with a flow-matching objective (Lipman et al., 2023) instead of the DDPM. In the follow-up work, unconditional FrameFlow was complemented with motif-guidance (loss-guidance) and motif amortisation, and was shown to beat RFdiffusion for some motif-scaffolding targets. The focus of the aforementioned works was structure conditioning, and neither of them has considered dynamical properties. Eigenfold (Jing et al., 2023) incorporates the physical constraints for oscillations into the diffusion SDE, but it has not been shown to influence downstream dynamical protein properties. Including coarse-grained NMA information in the conditioning is the most recent direction firstly proposed in (Komorowska et al., 2024). This work uses a loss-guidance formulation to condition on the lowest normal mode, but in contrast to our methodology, it does not have a trainable component, and conditioning efficiency suffers from the approximations needed for the loss-guidance term calculation. Also, Komorowska et al. (2024) show how one can implement two loss-guidance components at the same time, one for dynamics, and second for the motif scaffolding. Since RFdiffusion was trained to perform motif scaffolding if the motif information is provided, we have no need to include loss-guidance for structure at inference, and we avoid fine-tuning two guidance scales at the same time at inference.

**RFdiffusion** (Watson et al., 2023) is one of the leading diffusion models for protein design, trained to perform both unconditional and structure conditioned generation, and we use it as the dynamics unconditional base model in this work. It operates on $[N, C_\alpha, C]$ representation of the protein backbone. Atom coordinates are derived from two components: the translations of $C_\alpha$ atoms from the origin, and residues' orientation frames that position $N$ and $C$ atoms w.r.t. $C_\alpha$. Because of this division, the diffusion is decomposed into

two SDEs as well, one purely on the Euclidean space of $C_\alpha$ translations, and the second on $SO(3)$ space of frame rotations. The generation starts from fully noised $x_T$, and at each denoising step **RFdiffusion directly predicts the fully denoised sample** $\hat{x}_0$, which makes it particularly well suited for the approximation in the Equation 3. In the task of motif scaffolding, the information about motif target is explicitly contained in the noisy sample $x_t$; motif $C_\alpha$ atoms and frames are set to the ground truth and never noised.

## 3. NMA-tune

We tackle the following joint conditioning problem: can we design a protein backbone consisting of heavy atoms that will have a pre-defined structural motif, and motif $C_\alpha$ atoms will move in the lowest non-trivial normal mode according to the pre-defined displacement vectors?

**Methodology.** NMA-tune relies on finding a good approximation to the guidance term $\nabla_{x_t} \ln p(y|x_t)$ in Equation 4 using the neural network. We still take advantage of the insight that a well-chosen loss function allows to draw a probabilistic connection between $x$ and $y$ in cases when a neural network will be unable to do so. Here, $y$ is the eigenvector of the Hessian of the potential function. As pointed out in (Komorowska et al., 2024), there are currently no neural network architectures able to calculate eigenvectors of a matrix and to broadly generalise to weakly restricted set of unseen matrices.[1] Our key innovation is that we take this idea further and we improve the dynamics conditioning by: 1) replacing $\nabla_{x_t} l_y(\hat{x}_0)$ with $\nabla_{\hat{x}_0} l_y(\hat{x}_0)$ which improves sampling speed; and 2) passing this term to a trainable conditioner that finds a better conditional term than by directly using the loss-guidance. The conditioner receives both $\nabla_{\hat{x}_0} l_y(\hat{x}_0)$ and the unconditional score, and learns the best correction to the unconditional term along the path specified by the loss.

**Notation.** We denote the 3D coordinates of $[N, C_\alpha, C]$ backbone consisting of the $L$ residues by the matrix $x \in \mathbb{R}^{3L \times 3}$. From $L$ residues, $M$ residues constitue the motif enforced to have a given atomic placement, and the remaining residues are the scaffold such that $x = x_M \cup x_S$. The function $v_M$ calculates, for a given $x$, the $C_\alpha$ displacement vectors of $M$ residues in the lowest non-trivial mode of oscillation, and the output has the shape $v_M(x) \in \mathbb{R}^{M \times 3}$. The target displacements of $M$ residues are arranged in the matrix $y \in \mathbb{R}^{M \times 3}$. We denote the noise prediction calculated from RFdiffusion output $\hat{x}_0$ as $\epsilon_{RF}$, and $\epsilon_\theta$ is the

noise correction from the dynamics conditioner. The final noise prediction is a sum of the RFdiffusion noise and the conditioner correction

$$\epsilon = \epsilon_{RF} + \epsilon_\theta, \tag{5}$$

where we freeze the parameters of $\epsilon_{RF}$ during training.

### 3.1. Loss-guidance term calculation

With the ideal choice of $l_y(x)$, backpropagating through $\nabla_{x_t} l_y(\hat{x}_0)$ should give the desired guidance term. However, our goal is to find the ideal $\nabla_{x_t} l_y(\hat{x}_0)$ with the help of the trainable conditioner $\epsilon_\theta$. For this reason, we can avoid computationally expensive backpropagation through the RFdiffusion to obtain $\nabla_{\hat{x}_t} l_y(\hat{x}_0)$. Instead, we calculate cheaper $\nabla_{\hat{x}_0} l_y(\hat{x}_0)$: this quantity describes how the residues of $\hat{x}_0$ should reposition to minimise the loss. Note that

$$\nabla_{x_t} l_y(\hat{x}_0) = \frac{\partial \hat{x}_0}{\partial x_t} \nabla_{\hat{x}_0} l_y(\hat{x}_0), \quad \text{and} \tag{6}$$

$$\frac{\partial \hat{x}_0}{\partial x_t} = \frac{1}{\sqrt{\bar{\alpha}_t}} (I - \sqrt{1 - \bar{\alpha}_t} \frac{\partial \epsilon}{\partial x_t}). \tag{7}$$

Therefore, the expensive $\nabla_{x_t} l_y(\hat{x}_0)$ is equal to cheap $\nabla_{\hat{x}_0} l_y(\hat{x}_0)$ multiplied by the term dependent on the Jacobian $\frac{\partial \epsilon}{\partial x_t}$. We empirically found that ignoring the Jacobian and passing $\nabla_{\hat{x}_0} l_y(\hat{x}_0)$ as input to the conditioner yields satisfactory results. Similar observations about neglecting the Jacobian term were made by Poole et al. (2023). As detailed in Equation 5, we keep the parameters of the RFdiffusion network fixed and only train a substantially smaller correction network that takes $\nabla_{\hat{x}_0} l_y(\hat{x}_0)$ as an additional context.

### 3.2. Dynamics loss for guidance

The condition $y$ enforces that the selected subset of residues should have user-specified relative displacement directions and amplitudes. Therefore, the loss function should be invariant to the protein rotations and its length. Note that the presence of the motif introduces the frame of reference, hence the need to use modified version of NMA-loss rather than its first formulation from (Komorowska et al., 2024). Modified NMA-loss is:

$$l_y(x) = 1 - \cos(\tilde{y}, \tilde{v}_M(x))^2 \tag{8}$$

where $\tilde{y}$ and $\tilde{v}_M(x)$ are matrices $y$ and $v_M(x)$ flattened into 1D vectors and normalised. If $\tilde{y}$ and $\tilde{v}_M(x)$ have identical entries (up to the eigenvector sign), the motif residues move with the target amplitudes and angles with respect to each other. $v_M(x)$ uses the in-built PyTorch function to calculate matrix eigenvectors. The Hessian matrix was calculated using the Biotite (Kunzmann & Hamacher, 2018) implementation of the Hinsen (Hinsen & Kneller, 1999) force-field with cutoff radius 13Å, which is a coarse-grained model of

---

[1]Spectre (Martinkus et al., 2022) shows a way to generate a graph from Laplacian eigenvectors where the neural network must exhibit an understanding of the eigenvectors. However, eigenvectors of the Laplacian depend only on the graph edge weights, while the dynamics conditioning problem requires calculating the second order derivatives as well.

$C_\alpha$ interactions, and ignores the presence of other atoms. All operations described above allow for backpropagation when taking $\nabla_{\hat{x}_0}$.

$y$ is extracted from a protein that contains the motif substructure, and the orientation of $y$ w.r.t. the origin depends on the motif orientation. For each $\hat{x}_0$, we find the best rotation matrix $R$ of the target motif $C_\alpha$ coordinates to $\hat{x}_{M,0}$ using Kabsch alignment (Kabsch, 1976; 1978), and calculate the loss only after applying $R$ to $y$. This ensures that the loss is invariant to protein rotations, and the displacements' orientations w.r.t. motif residues orientations should match the target as well.

## 4. NMA-tune conditioner implementation and training

### 4.1. Architecture and inputs

$\epsilon_\theta$ is a rotation-equivariant Geometric Vectron Perceptron-Graph Neural Network (GVP-GNN) with five layers, which operates on vector and scalar features of nodes and edges. Node vector features are normalised vectors $\nabla_{\hat{x}_0} l_y(\hat{x}_0)$ and normalised unconditional noises $\epsilon_{RF}$. Node scalar features are the magnitudes of vectors $\nabla_{\hat{x}_0} l_y(\hat{x}_0)$, the magnitudes of $\epsilon_{RF}$, residue index along the backbone, and timestep $t$. Edge features are the displacement vectors between $C_\alpha$ atoms. Note that while RFdiffusion operates on $[N, C_\alpha, C]$ representation, GVP operates on a fully-connected graph of $C_\alpha$ atoms only, and it corrects the translational component of the RFdiffusion noise without modifying the rotational part. Workflow and architecture details are presented in Figure 6 in Appendix B. **Whenever we say that the denoiser takes as input scalar or vector features of $x$, we mean $C_\alpha$ features selected from $[N, C_\alpha, C]$ backbone features**. GVP does not add correction to the noise added to $x_{M,t}$ in order to avoid counteracting structure conditioning.

### 4.2. Training

The corrector $\epsilon_\theta$ is trained with the noise-matching objective and a number of auxiliary losses. In a single train step, $[N, C_\alpha, C]$ backbone is noised to time $t$ with the default RFdiffusion noise schedule and $T_{max} = 50$. RFdiffusion makes a prediction of $\hat{x}_0$, from which we get the dynamics-unconditional noise with motif scaffolding. For each protein, we randomly choose a motif that is kept constant in $x_t$ with length uniformly sampled between 10-40 residues. Motifs longer than 25 residues are made discontinuous, while the ones equal to 25 residues and shorter are made to have 50% chance to be discontinuous. Timesteps of motif residues are set to 0 for both RFdiffusion and $\epsilon_\theta$. Similarly, motif $\nabla_{\hat{x}_0} l_y(\hat{x}_0)$ and $\epsilon_{RF}$ vectors are set to 0 when passed to the conditioner. With these inputs, the conditioner

makes the prediction of the conditional noise correction, which is then summed with $\epsilon_{RF}$.

The key part of $\epsilon_\theta$ training is the minimisation of the noise-matching loss on the $C_\alpha$ translations. In this way the conditioner learns the valid protein structure, and to assure it is also dynamics-aware, we minimise the NMA-loss. In order to avoid the undesired effects of conditioning pointed out in (Komorowska et al., 2024), such as unrealistic chain distances or radii of gyration, we introduce auxiliary losses to enhance training and weight them appropriately. We denote the expected position at $t = 0$, calculated using the corrected noise $\epsilon$, by $\hat{x}_{0,corr}$, original sample by $x_0$, chain distances between $C_\alpha$ in $x$ by $d_{C_\alpha}(x)$, and $RG(x)$ is the radius of gyration calculated using $C_\alpha$ of $x$. We define the chain loss as

$$L_{\text{chain}} = \frac{MSE(d_{C_\alpha}(\hat{x}_{0,corr}), d_{C_\alpha}(x_0))}{MSE(d_{C_\alpha}(\hat{x}_0), d_{C_\alpha}(x_0))} \quad (9)$$

and the radius of gyration loss as

$$L_{\text{rg}} = |RG(\hat{x}_{0,corr}) - RG(\hat{x}_0)|. \quad (10)$$

$L_{\text{chain}}$ and $L_{\text{rg}}$ ensure that large scale properties of conditioned backbones do not deviate from RFdiffusion predictions. All losses are weighted as

$$L = 0.05 * L_{\text{noise}} + 0.8 * L_{\text{NMA}} + 0.1 * L_{\text{chain}} + 0.05 * L_{\text{rg}} \quad (11)$$

where $L_{\text{NMA}}$ is the same as $l_y(x)$ in Equation 8. $\epsilon_\theta$ is trained for 10 epochs with Adam optimiser, which took $\approx 15h$ on a single Nvidia A100 80GB. The learning rate 1e-4 is decreased by 0.1 after 5000 gradient updates and batch size is 32. The $L_{\text{noise}}$ term is the standard denoising loss in the NMA-tune parametrisation $L_{\text{noise}} = \mathbb{E}_{t,y,x_0,\epsilon} ||\epsilon - (\epsilon_{RF}(\sqrt{\overline{\alpha}_t}x_0 + \sqrt{1-\overline{\alpha}_t}\epsilon) + \epsilon_\theta(\sqrt{\overline{\alpha}_t}x_0 + \sqrt{1-\overline{\alpha}_t}\epsilon, y))||^2$.

**Dataset** for training $\epsilon_\theta$ was based on the SCOPe database (Fox et al., 2013; Chandonia et al., 2021). We use the data preprocessing from Genie (Lin & AlQuraishi, 2023) to remove proteins with multiple chains or missing atoms, and to assure no two domains share >40% sequence identity. Additionally, we filter out proteins shorter than 50 or longer than 256 residues. 7139 proteins remained and we used train:validation:test split 0.8:0.1:0.1.

**Structure and dynamics targets** We performed a literature search for proteins with well investigated protein hinge motions and prioritised having a smaller set of high quality targets over a larger set but less carefully verified. We chose: triglyceride lipase (Derewenda et al., 1992) (PDB id: 4tgl), calmodulin (Khade et al., 2021) (PDB id: 1exr), and HIV-1 protease in semi-open conformation (Hornak et al., 2006) (PDB id: 1hhp). We extract four targets from those, which we call after the protein they come from, that is, 4tgl, 1exr, 1hhp for a single asymmetric unit of HIV-1 protease, and 1hhp_assembly for protease biological assembly (more discussion of the targets is in the Appendix D).

We used PACKMAN software (Khade et al., 2020) to verify that the hinge motion described in the literature is found at our targets. 1hhp_assembly target, which we derive from two flexible flaps of two symmetrically positioned chains, is the only discontinuous target that we use. We did not use targets from (Komorowska et al., 2024), since those were shown to have low designability scores even without dynamics conditioning, thus preventing the generation of designable dynamics-conditioned proteins. We focus on hinge motions since they are the flagship example of importance of flexibility for function, and also NMA-guidance was originally evaluated on hinge targets and we aim for a comparable benchmark. Note that NMA-tune can in theory be applied to any collective type of motion.

**Evaluation metrics** We evaluate the samples with respect to the dynamics and designability. We measure the success of the dynamics conditioning in terms of cosine similarity (cossim) of vectors $\tilde{y}$ and $\tilde{v}_M(x)$. Designability metric answers the question: for a given backbone structure, is there a sequence that will fold into it? The metric is computed using the standard pipeline of inverse-folding and folding again, commonly used in other works (Yim et al., 2023b; Lin & AlQuraishi, 2023; Yim et al., 2023a), and we explain it here again for consistency. We follow the RFdiffusion steps to compute the oxygen positions in the RFdiffusion $[N, C_\alpha, C]$ generated backbone. For each such backbone, we generate eight amino-acid sequences using ProteinMPNN (Dauparas et al., 2022) at sampling temperature 0.1, and we use the version of the model that uses all four atoms instead of $C_\alpha$ only. The target motif amino-acids are known and also passed to ProteinMPNN. Those sequences are then folded with ESMFold (Lin et al., 2022) to create the "ESMFold designs". For each sample we choose one design that has the lowest RMSD to the original backbone, which is called the self-consistency RMSD (scRMSD). Additionally, we compute the self-consistency RMSD of motif residues only, which is more informative about the success of the motif placement than the global scRMSD. Samples with scRMSD $< 2$ and sc-motif-RMSD $< 1$, and whose chosen designs have pLDDT $> 0.7$ and pAE $< 10$, are deemed designable. For calculations of both scRMSD and sc-motif-RMSD, we use $C_\alpha$ representation. In the end, we compute cossims between target displacements and displacements in: 1) RFdiffusion $C_\alpha$ backbone output (orig-cossim); and 2) ESMFold designed $C_\alpha$ backbone (self-consistency-cossim, or sc-cossim). The latter amounts to performing NMA again on the ESM design using the same force-field parametrisation as utilised in the conditioning. Since ESMFold designs are a more faithful representation of the protein that will be synthesised than the direct RFdiffusion outputs, sc-cossim quantifies whether the conditioning effects can be observed in the real-world proteins. And finally, as the last evaluation step, we select samples of

high structure quality to assess whether the targeted NMA corresponds to the oscillation mode in the MD trajectory. We give detailed description of this procedure in Section 5.2.

# 5. Results

## 5.1. Conditioning efficiency

Empirically, we found that upscaling $\epsilon_\theta$ term was necessary to observe the effect of conditioning (sampling details and ablations are in the Appendix A). In RFdiffusion, but also in other works in the field (Yim et al., 2023b; Lin & AlQuraishi, 2023; Trippe et al., 2023), the diversity vs quality trade-off is controlled by downscaling the Gaussian noise term at inference time by a parameter $\eta \in [0, 1]$. Smaller $\eta$ can potentially improve the sample quality but at the cost of diversity. Since $\eta$ was an important part of the original RFdiffusion evaluation, we ablate how NMA-tune behaves for 3 $\eta$ values and benchmark against NMA-guidance. We reimplement and fine-tune NMA-guidance as a part of RFdiffusion, so any difference in performance between the two methods cannot be attributed to the base unconditional model. Table 1 shows the comparison between structure-only (column *dynamics-cond: none*) and jointly conditioned samples (columns *dynamics-cond: tune* and *dynamics-cond: guid.*). Note that structure-only samples set the upper bound on designability achievable for joint conditioning. Targets 1exr and 1hhp have high designability in both conditioning scenarios, but 4tgl and 1hhp_assembly were not possible to scaffold even with structure conditioning only. For the latter two targets, we changed the designability threshold of sc-motif-RMSD to $< 3$Å and $< 2$Å respectively. However, we still find these results informative, as they show that: 1) dynamics conditioning can be successful even for difficult structural targets; and 2) dynamics conditioning can work well for future models with improved motif scaffolding capabilities. For each target we sampled scaffold of different lengths. To ensure that differences in sc-cossim between NMA-tune and NMA-guidance are not caused by the differences in scaffold lengths, we seeded generation of each batch of 110 samples, and repeated the procedure for three random seeds. Table 1 shows the mean over three seeds, and the results for the individual runs are in the Appendix E.

Joint conditioning improves sc-cossim score across different targets, and our method consistently outperforms NMA-guidance. Since dynamics conditioning is a very novel direction in the generative protein modelling, there are no other methods to compare ours other than NMA-guidance. **When we report the fraction of samples with sc-cossim above some threshold, we mean the fraction of all samples that are designable and have desired sc-cossim**. Both NMA-guidance and NMA-tune sacrifice designability to achieve conditioning. However, evaluation of simple structural metrics such as $C_\alpha$ backbone distance, radius of gyration and secondary structure compo-

*Table 1.* Success rates for NMA-tune (tune) and NMA-guidance (guid.) evaluated on 4 targets and ablated with respect to noise scale $\eta$. For each datapoint we take 110 samples. We average the results over 3 random seeds. Even though NMA-guidance produces more designable samples overall, NMA-tune beats NMA-guidance in terms of successful structure and dynamics conditioned designable samples. *Designability for 4tgl and 1hhp_assembly is calculated according to the changed sc-motif-RMSD criterion. † stands for 1hhp_assembly.

$\eta = 0.0$

| Target | 1hhp | | | 1exr | | | 1hhp_a† | | | 4tgl | | |
| Dynamics-cond | tune | guid. | none | tune | guid. | none | tune | guid. | none | tune | guid. | none |
| --- | --- | --- | --- | --- | --- | --- | --- | --- | --- | --- | --- | --- |
| % designable* | 58.2 | 67.3 | 72.7 | 53.3 | 68.8 | 81.8 | 20.3 | 47.7 | 51.8 | 11.5 | 13.2 | 13.0 |
| % w. sc-cossim $> 0.9$ | **12.4** | 7.6 | 1.2 | **9.4** | 6.4 | 1.2 | **1.2** | 0.6 | 0.0 | 0.0 | 0.0 | 0.0 |
| % w. sc-cossim $> 0.8$ | **20.3** | 15.5 | 7.3 | **26.1** | 20.9 | 4.8 | **7.0** | 2.1 | 0.6 | **0.9** | 0.0 | 0.0 |
| % w. sc-cossim $> 0.7$ | **29.1** | 22.7 | 11.8 | **33.0** | 30.0 | 11.5 | **9.1** | 5.2 | 1.2 | **2.4** | 1.5 | 0.3 |

$\eta = 0.1$

| Target | 1hhp | | | 1exr | | | 1hhp_a† | | | 4tgl | | |
| Dynamics-cond | tune | guid. | none | tune | guid. | none | tune | guid. | none | tune | guid. | none |
| --- | --- | --- | --- | --- | --- | --- | --- | --- | --- | --- | --- | --- |
| % designable* | 53.6 | 63.0 | 70.3 | 53.9 | 67.6 | 81.2 | 17.9 | 49.5 | 54.5 | 10.9 | 12.7 | 13.9 |
| % w. sc-cossim $> 0.9$ | **10.9** | 6.7 | 3.0 | **10.0** | 5.8 | 0.6 | **1.5** | 0.3 | 0.0 | 0.0 | 0.0 | 0.0 |
| % w. sc-cossim $> 0.8$ | **20.3** | 13.3 | 10.0 | **22.4** | 21.5 | 7.0 | **4.2** | 1.5 | 1.5 | 0.0 | **0.3** | 0.0 |
| % w. sc-cossim $> 0.7$ | **26.7** | 17.9 | 15.2 | **32.1** | 30.9 | 11.8 | **8.2** | 5.5 | 2.4 | **0.6** | 0.3 | 0.3 |

$\eta = 1.0$

| Target | 1hhp | | | 1exr | | | 1hhp_a† | | | 4tgl | | |
| Dynamics-cond | tune | guid. | none | tune | guid. | none | tune | guid. | none | tune | guid. | none |
| --- | --- | --- | --- | --- | --- | --- | --- | --- | --- | --- | --- | --- |
| % designable* | 42.7 | 58.5 | 57.3 | 46.4 | 59.7 | 68.2 | 15.8 | 36.7 | 37.6 | 8.2 | 13.9 | 13.3 |
| % w. sc-cossim $> 0.9$ | **6.1** | 3.0 | 0.9 | **4.8** | 0.9 | 0.3 | **0.3** | 0.0 | **0.3** | 0.0 | 0.0 | 0.0 |
| % w. sc-cossim $> 0.8$ | **13.9** | 11.5 | 6.1 | **17.3** | 9.7 | 1.8 | **2.4** | 0.9 | 0.3 | **0.3** | **0.3** | 0.0 |
| % w. sc-cossim $> 0.7$ | **20.9** | 16.1 | 10.9 | **25.8** | 19.7 | 6.4 | **5.8** | 2.7 | 2.7 | **1.2** | 0.9 | 0.6 |

sition does not reveal significant differences between them. We show structural sanity checks for sets of all samples and for samples filtered for designability in Appendix B. Still, NMA-tune wins in terms of designability and dynamics-conditioning at the same time, which is the only subset of samples that can be counted as successful designs. **It is not enough to obtain a high quality sample - a sample must be both designable and meet the dynamics conditions, and NMA-tune offers best designability/conditioning strength trade-off.**

Since force-field Hessian depends on the pair-wise distances, modes of oscillation will be related to the structure, and structure conditioned samples will match the dynamics targets to some degree. For example, a small fraction of structure conditioned samples for 1exr target has high sc-cossim $> 0.9$. Still, jointly conditioned samples show improved match to the dynamics target as compared to structure conditioned only, as is in the case of 1exr and $\eta = 0.0$ where the fraction of samples with best sc-cossim increases about 8 times with NMA-tune, and 5 times with NMA-guidance. And for some cases, as for 1exr and $\eta = 1.0$, generating the highest sc-cossim samples is possible only with the dynamics conditioning present. As expected, the designability can go down as $\eta$ increases for some targets, and dynamics conditioning has a negative impact on the designability. Despite this undesired effect, the percentage of samples that are designable and exhibit target dynamics is higher when conditioning jointly, which means more efficient conditional sampling overall. Figure 7 in Appendix B shows the distributional shift of sc-cossims to higher values for designable, dynamics-conditioned samples.

Additionally, our method not only increases the proportion of effectively conditioned samples as compared to NMA-guidance, but also offers a significant improvements of a sampling speed. For example, on a single Nvidia A100 80GB, our method gives much faster sampling of a single protein (note that official implementation of RFdiffusion is compatible only with batch size 1). NMA-tune increases the sampling time only by 8% as compared to sampling without dynamics conditioning, while NMA-guidance causes the increase by about 75%.

To gain an intuitive understanding of a good and a bad match to the targeted normal mode, we show in Figure 1 the visualisations of two samples, one with high sc-cossim and another with low sc-cossim.

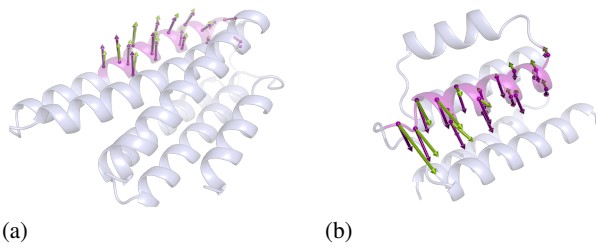

(a)           (b)

*Figure 1.* Examples of **(a)** a design that matches the targeted normal mode (sc-cossim= 0.96), and **(b)** one that shows poorer match (sc-cossim= 0.68). Both visualisations show the ESMFold designs for the RFdiffusion backbones. Green arrows show the displacements of the targeted $C_\alpha$ atoms in the ESMFold designs, and purple show the target displacement (arrows are upscaled for clarity). Greater sc-cossim assures that the relative amplitudes and angles of the green displacement vectors reflect amplitudes and angles between purple displacement vectors.

### 5.2. MD evaluation

As a final step in our evaluation, we checked whether the targeted normal mode is observed in the MD simulation of the successfully conditioned sample. To this end, we chose 4 ESMFold designs that correspond to designable samples, two with high and two with low sc-cossim, all for 1hhp target. Since the way in which protein explores available conformations is inherently stochastic, we run three independent 500 ns simulations (replicas) per ESMFold design. **The following simulations show that short-term motions of dynamics-conditioned samples are consistent with the targeted mode.** In the next paragraphs, we explain the evaluation protocol and relevant metrics.

We perform PCA on the generated trajectories of $C_\alpha$ atoms and extract the largest principal component. Similarly, as with sc-cossim evaluated on the ESMFold design, we compute the cosine similarity between the displacement vectors of motif residues in the PCA mode and in the targeted mode – we call this metric PCA-cossim. Note that due to the probabilistic nature of MD simulations, proteins may explore less likely motions, even if the targeted normal mode corresponds to the most probable collective motion. Since there is never a guarantee that the most likely domain motion will occur in a given single replica, we report the best PCA-cossim value from the three replicas for each sample.

Two designs were chosen from successfully dynamics-conditioned NMA-tune samples, and the other two from dynamics-unconditional samples. We selected designs with low MolProbity score (as assessed by SwissModel service (Waterhouse et al., 2024)): dynamics-conditioned with sc-cossim 0.92 and 0.97 and MolProbity (Williams et al., 2018) scores 1.68 and 1.91, and dynamics-unconditional with sc-cossim 0.16 and 0.32 and MolProbity scores 1.52 and 1.69. For intuitive explanation of MolProbity see Ap-

*Table 2.* Values of PCA-cossim at 20-100 ns interval for NMA-tune dynamics-conditioned and dynamics-unconditional samples. PCA-cossim is shifted to higher values for the dynamics-conditioned samples.

|  | dynamics-cond. | | dynamics-uncond. | |
|---|---|---|---|---|
| sc-cossim | 0.92 | 0.97 | 0.16 | 0.32 |
| PCA-cossim | 0.86 | 0.58 | 0.23 | 0.34 |

*Table 3.* Values of PCA-cossim computed using the tailored time interval (durations in Table 7 in Appendix C).

|  | dynamics-cond. | | dynamics-uncond. | |
|---|---|---|---|---|
| sc-cossim | 0.92 | 0.97 | 0.16 | 0.32 |
| PCA-cossim (interval) | 0.50 | 0.48 | 0.51 | 0.80 |

pendix C. Before running the simulations, the ESMFold designs were minimised and equilibrated, with $C_\alpha$ RMSDs to their initial structures remaining below 0.5 Å. A detailed description of the simulation protocol and the subsequent evaluations can be found in Appendix C.

Initially, PCA-cossim values were analysed during the early simulation period (20–100 ns), once the proteins had equilibrated but had not significantly deviated from their starting structures. Table 2 shows the results: the first two columns are for dynamics-conditioned samples and the last two columns are for dynamics-unconditional samples. The PCA-cossim values are higher for dynamics-conditioned samples, indicating a higher overlap between the PCA mode and the targeted mode. While improvements in PCA-cossim due to dynamics conditioning are not as pronounced as improvements in sc-cossim, they remain significant. Dynamics conditioning remains effective in the short-term simulated motions, which is promising for the potential downstream applications.

Finally, we analysed the presence of the targeted mode during the later stages of the simulation, once the proteins had enough time to sample alternative conformations. To this end, we find the stable temporal regions by plotting $C_\alpha$ RMSD vs time (Figure 8 in Appendix C), and perform PCA on the trajectories from stable time intervals. As expected, PCA-cossim is lower as compared to initial time intervals (Table 3). While it is unsurprising that the direction of the lowest oscillation mode differs between alternative conformations, the ideal scenario would involve the targeted normal mode persisting in the new stable states reached during the simulation. Together with the early PCA-cossim results, these findings suggest that the initial motions of dynamics-conditioned samples better align with the targeted mode, indicating successful conditioning of the starting conformations.

### 5.3. Limitations

There is room for improvement in terms of stability of the designed proteins. The targeted mode does not prevail for alternative states in the simulation in a consistent way, but it still might occur spontaneously (sample sc-cossim=0.32).

Like NMA-guidance, NMA-tune takes the displacements' directions into account, which does not allow for a direct control of the absolute flexibility in the conditioned region. Important direction for further work is to extend conditioning to the absolute values of displacement vectors as well.

## 6. Conclusions

In this work we present NMA-tune, the methodology for protein generative modelling conditioned on the normal mode of oscillation. In contrast to existing techniques, NMA-tune utilises RFdiffusion and benefits from the trainable conditioner that allows sampling along the path specified by loss-guided diffusion. NMA-tune outperforms existing techniques as it produces designable samples with higher dynamics conditioning success rate.

## Impact Statement

This paper presents work whose goal is to advance the general methodology of generative protein design. The potential consequences of our work for medicine and chemistry are very broad and hard to quantify, and there is nothing specific we are able to highlight here.

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

# Appendices

## A. Sampling details

We used 50 sampling steps, which is the default in RFdiffusion. We observed it was necessary to upscale the conditional term by some guidance scale to obtain samples both designable and with improved dynamics. We obtained best results if introduced the sinusoidal time dependence for the guidance scale $gs$, such that $gs = gs_{const} * \sin t * \pi$, where in the sampling the normalised time flows from 1 to 0. Firstly, we set $\eta = 0.0$ and then searched over several $gs_{const}$. We did not add dynamics conditioning correction in the last 3 generation steps. Then, the best guidance scales for each target were used for experiments with $\eta = 0.1$ and $\eta = 1.0$ (the best guidance scale in a sense of the largest proportion of designable samples with sccosim >0.9). We searched over $gs_{const} \in 20, 40, 60, 80, 100$ and additionally 120 for 1hhp_assembly, and we chose $gs_{const} = 80$ for all targets expect 1hhp_assembly for which $gs_{const} = 100$. Additionally, for targets 1hhp and 1exr we ablate the time scaling function in Table 4.

*Table 4.* NMA-tune ablation with respect to the guidance strength time scaling in the sampling. Linear corresponds to $gs = gs_{const} * t$, where time flows from 1 to 0, and constant is simply unscaled $gs_{const}$. Linear and constant scalings perform similar to sinusoidal scaling from the main text, and still outperform NMA-guidance. We also searched for $gs_{const}$ with finer interval. Values of $gs_{const}$ chosen for each linear scaling and target were 70 for 1hhp and 30 for 1exr, and for constant scaling were 70 for 1hhp and 20 for 1exr.

| | $\eta = 0.0$ | | | | $\eta = 0.1$ | | | | $\eta = 1.0$ | | | |
|---|---|---|---|---|---|---|---|---|---|---|---|---|
| Target | 1hhp | | 1exr | | 1hhp | | 1exr | | 1hhp | | 1exr | |
| Time scaling | Linear | Constant | Linear | Constant | Linear | Constant | Linear | Constant | Linear | Constant | Linear | Constant |
| % designable | 59.39 | 38.18 | 63.64 | 64.55 | 55.76 | 31.52 | 62.42 | 68.79 | 39.70 | 27.58 | 61.21 | 57.88 |
| % w. sc-cossim > 0.9 | 11.21 | 9.09 | 11.52 | 11.82 | 7.88 | 10.00 | 7.58 | 12.42 | 5.15 | 5.76 | 6.06 | 7.27 |
| % w. sc-cossim > 0.8 | 19.39 | 16.67 | 29.39 | 32.12 | 17.27 | 14.85 | 25.76 | 33.64 | 13.33 | 11.21 | 18.48 | 18.79 |
| % w. sc-cossim > 0.7 | 27.58 | 21.21 | 38.18 | 40.91 | 23.94 | 19.09 | 37.58 | 47.27 | 17.58 | 14.24 | 29.39 | 29.39 |

In every single protein generation, we uniformly sampled length of the scaffold before (prefix) and after (suffix) the motif, between min and max values dependent on the target. For the discontinuous motif 1hhp_assembly we also sampled gap length between 20-50 residues.

We followed the steps in (Komorowska et al., 2024) to implement the benchmark method and also performed the $gs_{const}$ search. We evaluated the following values of $gs_{const}$ in 1000 intervals: 2000 to 9000 for 1exr and 1hhp; 2000 to 5000 for 1hhp_assembly; 2000 to 4000 for 4tgl. We chose 8000 for 1exr, 7000 for 1hhp, 4000 for 1hhp_assembly, 2000 for 4tgl. Similarly as in our method, we do not add the conditioning update in the last 3 steps.

For both methods we rescaled the corrected noise (which is the sum of the unconditional and conditional noise) such that it has the same variance as the unconditional noise, in order to avoid too large denoising steps and improve designability. However upon further investigation we found this effect was not affecting designability in a consistent way across different time scalings and guidance strengths. In the end, fine-tuning the guidance strength was the crucial factor that determined designability.

*Table 5.* Prefix and suffix ranges used in sampling.

| | Prefix min | Prefix max | Suffix min | Suffix max |
|---|---|---|---|---|
| 4tgl | 30 | 70 | 20 | 70 |
| 1exr | 30 | 70 | 20 | 70 |
| 1hhp | 30 | 50 | 30 | 50 |
| 1hhp_assembly | 30 | 50 | 30 | 50 |

Sampling was done on a single NVIDIA A100 80GB and the compute resources are comparable to only running RFdiffusion sampling.

## B. Extra figures

Figures 2 and 3 show mean $C_\alpha$ backbone distance, radius of gyration and secondary structure composition for samples $\eta = 0$ without filtering for sc-RMSD. For both NMA-tune and NMA-guidance, the $R_g$ distribution has a longer tail for the dynamics-conditioned samples as compared to dynamics-unconditional, however it remains in the physical range. Both methods do disturb secondary structure distribution, however the ratio helix:sheet:coil is preserved.

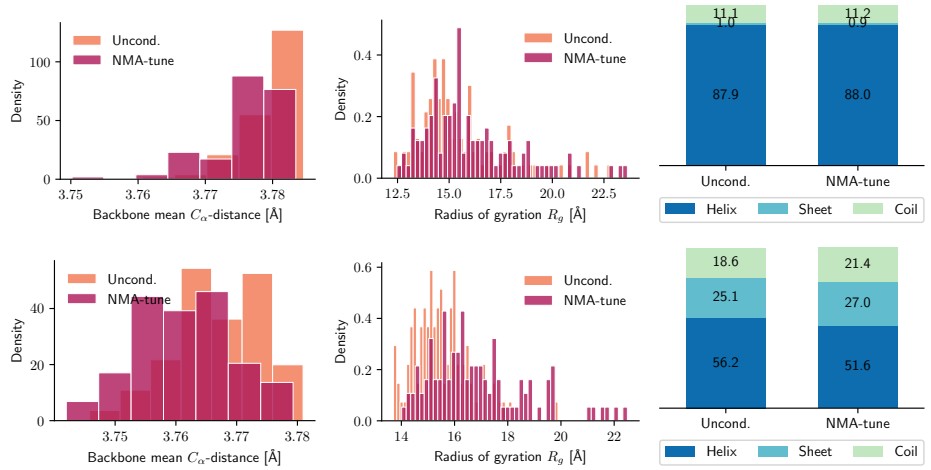

*Figure 2.* Structure metrics for NMA-tune. **Top** row: 1exr target. **Bottom** row: 1hhp_assembly target.

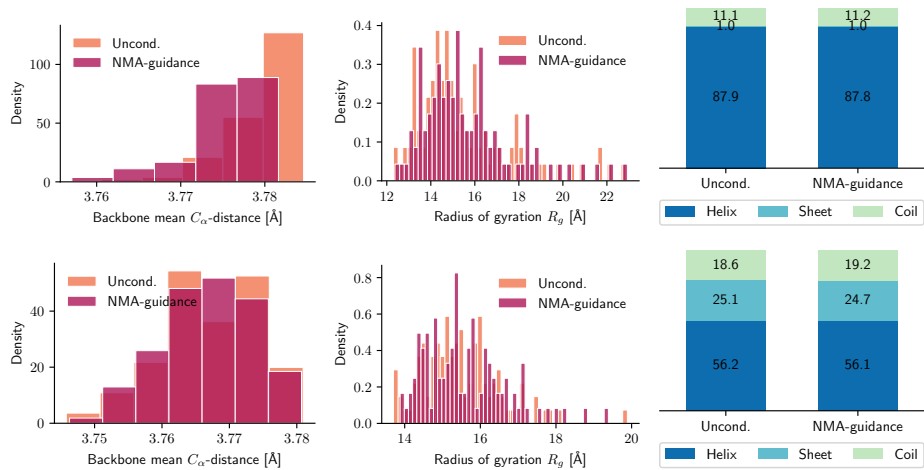

*Figure 3.* Structure metrics for NMA-guidance. **Top** row: 1exr target. **Bottom** row: 1hhp_assembly target.

Figures 4 and 5 show the same statistics as Figures 2 and 3, but for samples filtered for designability across all seeds. Statistics follow similar trend as when calculated over samples without filtering.

In the end, we perform a sanity check whether the differences in designability come from sampling more novel structures. To this end, to evaluate novelty and diversity for targets 1exr and 1hhp_assembly. For novelty, we compute the TM-score to AFDB and PDB100 databases available at Foldseek (van Kempen et al., 2024) server, and for each sample retain the max score. Table 6 shows mean values of those max TM-scores (the lower, the more novel samples). While it seems that NMA-tune outperforms NMA-guidance by a narrow margin, novelty of both methods remains in the range comparable to other generative models. We calculate diversity using MaxCluster (Herbert & Sternberg, 2008) with hierarchical clustering

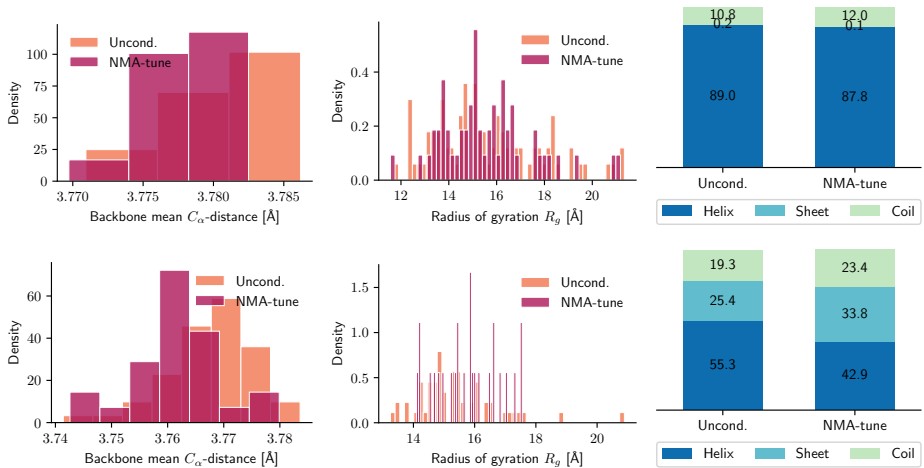

*Figure 4.* Structure metrics for NMA-tune (designable samples). **Top** row: 1exr target. **Bottom** row: 1hhp_assembly target.

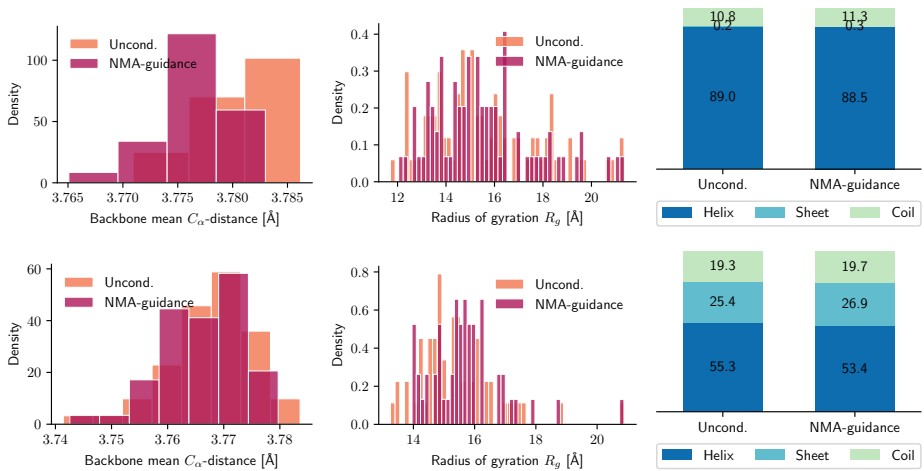

*Figure 5.* Structure metrics for NMA-guidance (designable samples). **Top** row: 1exr target. **Bottom** row: 1hhp_assembly target.

|            | 1exr, $\eta$=0.0 | 1exr, $\eta$=1.0 | 1hhp_a, $\eta$=0.0 | 1hhp_a, $\eta$=1.0 |
|------------|------|------|------|------|
| NMA-tune   | 0.68 | 0.64 | 0.52 | 0.53 |
| NMA-guid.  | 0.71 | 0.68 | 0.57 | 0.55 |

*Table 6.* Mean TM-score across different targets and noise levels.

(single linkage method), in sequence independent mode, with a TM-score threshold 0.6. From the set of all 110 samples generated per target per noise scale for one seed, we take samples for $\eta = 0.0$ and $\eta = 1.0$ together, and discard the non-designable samples. Results for the remaining designable samples are as follows (clusters / num of designable samples):
NMA-tune: 1exr target: 13/118; 1hhp_assembly target: 32/34
NMA-guidance: 1exr target: 10/137; 1hhp_assembly target: 52/91
As expected, diversity depends on the scaffolding target. Neither of the methods collapses to sample from a single cluster.

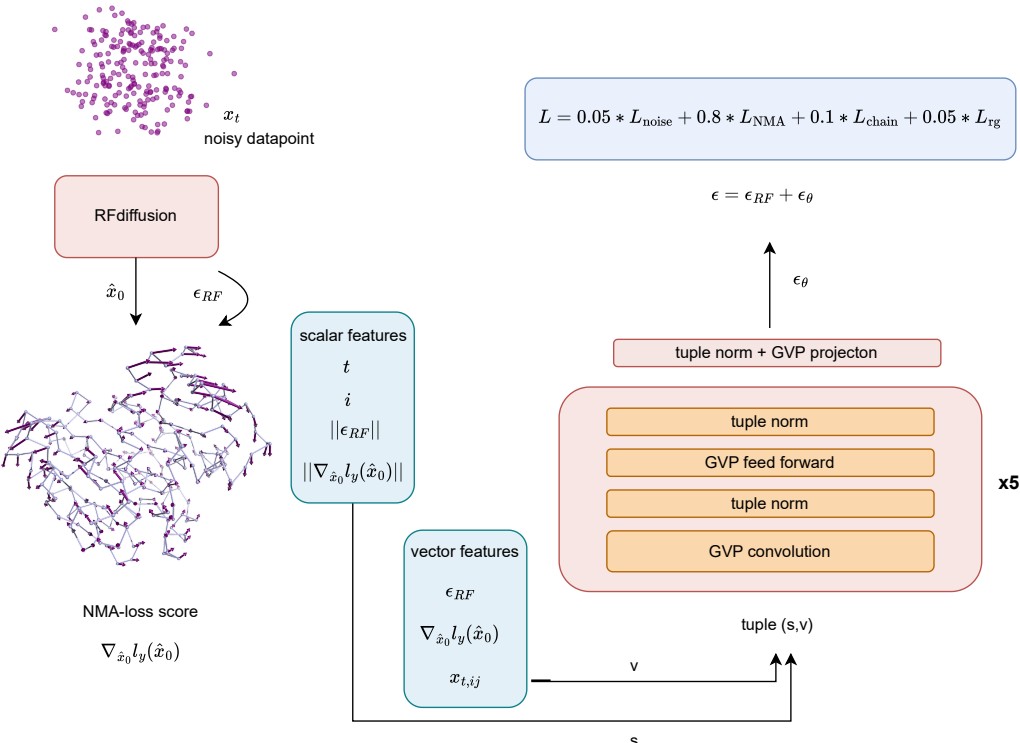

*Figure 6.* Conditioning framework diagram. The conditioner is a GVP based graph neural network, which operates on scalar (s) and vector (v) inputs. Firstly the tuple (s,v) are projected onto node and edge features. Then, inside the conditioner there are 5 stacks of graph convolution where the message passing function is the gated GVP, followed by the normalisation layer that normalises scalar and vector features, feedforward layer with a single gated GVP and another normalisation layer, which sums up to 530K parameters. In a training step, RFdiffusion takes noised $[N, C, C_\alpha]$ coordinates of the protein with a pre-selected structural motif and makes the dynamics-unconditional prediction of fully denoised sample $\hat{x}_0$. The dynamics-unconditional noise is derived from $\hat{x}_0$. The lowest oscillation mode in $\hat{x}_0$ is used to calculate the gradient of the NMA-loss. Next, the features are passed to the conditioner which performs message-passing on the graph of $C_\alpha$ atoms, and it outputs its correction.

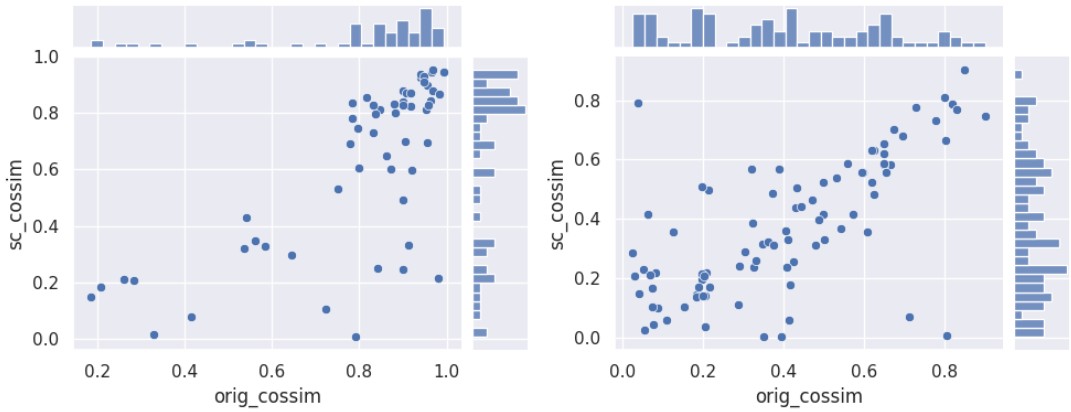

*Figure 7.* The plot of sc-cossim vs orig-cossim for conditional (left) and unconditional (right) samples for the 1exr target at ns=0.0. Conditional samples exhibit both higher orig-cossim and sc-cossim. Backbones with scRMSD=0 should lie on the line $y = x$.

## C. Extended MD discussion

### C.1. Scoring of sample quality

MolProbity is a validation tool for macromolecular 3D structures, such as proteins and nucleic acids. Its overall score is derived from a combination of geometric and stereochemical parameters (e.g., clash score, percentage of Ramachandran outliers, sidechain rotamers) that assess the quality of an atomic model. A lower MolProbity score indicates fewer geometric inconsistencies and generally denotes a higher quality structure, ensuring the accuracy of atomic coordinates. SwissModel is a homology modeling server that generates 3D protein models based on known structural templates and provides quantitative assessments of model quality.

### C.2. Simulation protocol

Molecular dynamics (MD) simulations were performed using GROMACS 2024 version and NVIDIA A100 80GB GPU. Simulations were carried out with Amber94 forcefield, in explicit solvent using TIP3P water models, with the system solvated in an octahedral box. Counter ions were added to neutralise the system, with a salt concentration of 150 mM. Periodic boundary conditions were applied to all replicas. The initial energy minimisation was performed using the steepest descent algorithm, with a maximum force tolerance of 10 kJ/mol/nm. A maximum of 50,000 steps was set to ensure system stabilisation. Following energy minimisation, the system was equilibrated in the NVT ensemble for 100 ps using a leap-frog integrator. The temperature was maintained at 300 K using a velocity-rescale thermostat. Next, the system was equilibrated in the NPT conditions for 100 ps under similar conditions: the pressure was maintained at 1 bar using a Parrinello-Rahman barostat. The simulations were performed in NPT conditions for a total of 500 ns each, with a time step of 2 fs. Coordinates and velocities were recorded at 10 ps intervals. All simulations were carried out with a cut-off distance of 1.0 nm for short-range electrostatic and van der Waals interactions. Long-range electrostatic interactions were treated with the Particle Mesh Ewald method. When selecting the appropriate samples for simulations, we verified the motif residues in the ESMfold are indeed flexible, and the L2 norm of concatenated displacements vectors of the motif residues were 0.31, 0.56, 0.90, 0.60 for samples with sc-cossim 0.97, 0.92, 0.32, 0.16 respectively. Figure 8 shows how $C_\alpha$ RMSD w.r.t the starting structure

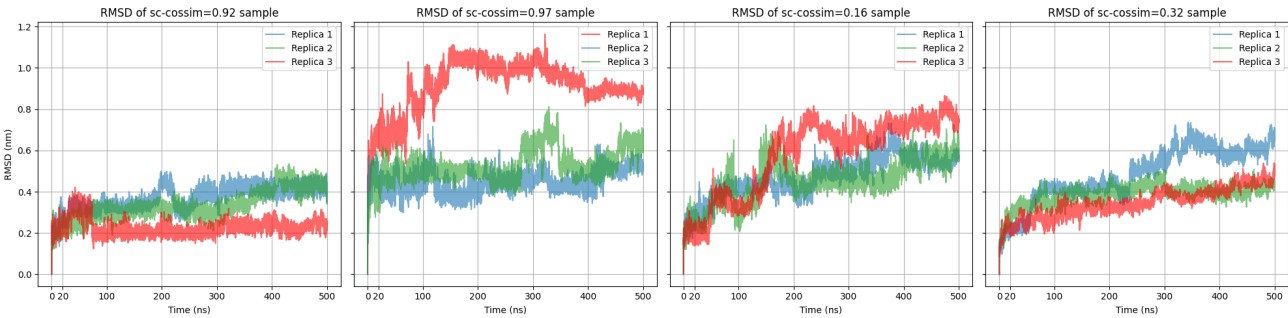

Figure 8. $C_\alpha$ RMSD vs. time during the simulations. Best-of-three replica, whose PCA-cossim is reported in Table 2, is plotted in red.

changes in time. Different replicas reach stable states in different time intervals, therefore for each one of them we choose the tailored time interval where RMSD flattens (Table 7). Even if a given replica does not seem to converge to a stable region for a longer time, we still try to find the best interval, and perform PCA for consistency.

Table 7. Time intervals of trajectories used to compute PCA-cossim in Table 3.

| sc-cossim | 0.92 | | | 0.16 | | | 0.97 | | | 0.32 | | |
|---|---|---|---|---|---|---|---|---|---|---|---|---|
| replica | 1 | 2 | 3 | 1 | 2 | 3 | 1 | 2 | 3 | 1 | 2 | 3 |
| interval start, end (ns) | 300, 500 | 100, 300 | 80, 500 | 225, 500 | 225, 400 | 225, 500 | 150, 300 | 150, 400 | 100, 225 | 100, 200 | 100, 250 | 100, 250 |

Computing PCA-cossim in all cases requires finding the rotation matrix $R$ (as in the NMA-loss in the Equation 8). $R$ aligns the frame of reference of the target displacements with the frame of the motif in the simulated sample. Since the motif residues are moving in the simulation, there is no one constant frame of reference. For this reason we align the target motif

residues onto the average positions of the motif residues in the sample, and the positions are averaged over any given time interval specified in text.

## D. Details of the hinge targets

**4tgl** (Figure 9) has been found to exhibit hinge motion of the lid residues 82-96. We found that choosing residues 70-89 as the motif resulted in protein close-to-designable. Additionally we used PACKMAN (Khade et al., 2020), a tool for finding protein hinges in the query structure, and we found presence of the hinge at residues 75-90.

Khade et al. (2021) found hinge motions at residues 62-87 of **1exr** (Figure 10). Choosing residues 64-81 as the motif resulted in the designable samples.

**1hhp and 1hhp_assembly** (Figures 11 and 12) are targets are extracted from flexible flaps of HIV-1 protease. Protease undergoes transformation between closed and open conformations with a pronounced movement of flaps (residues 43-58). We obtained two targets, one extracted from a single chain in the asymmetric unit, the other from a biological assembly with two chains. Residues for 1hhp target were 46-54, and for 1hhp_assembly target they were 46-50 and 145-149. For 1hhp_assembly, the exact hinge motion was found by PACKMAN not exactly at the flap, but at the spatially close residues that are placed near the flap in the gap between the two units in the assembly. We further confirmed that the target residues in 1hhp_assembly are indeed active in the lowest frequency motion using the elNemo software (Suhre & Sanejouand, 2004).

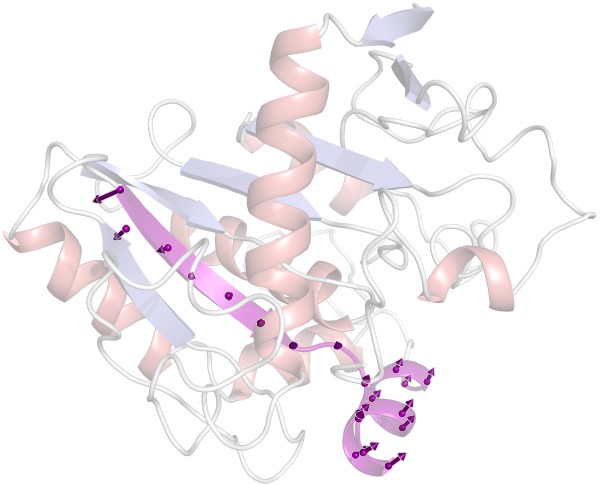

*Figure 9.* 4tgl target

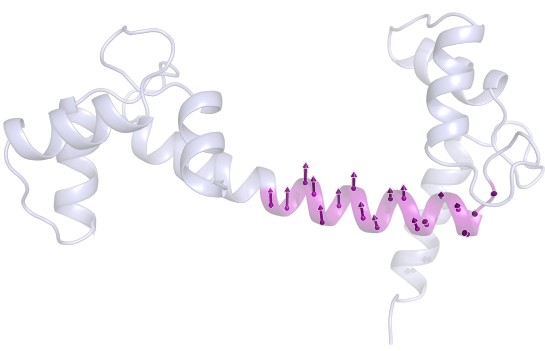

*Figure 10.* 1exr target

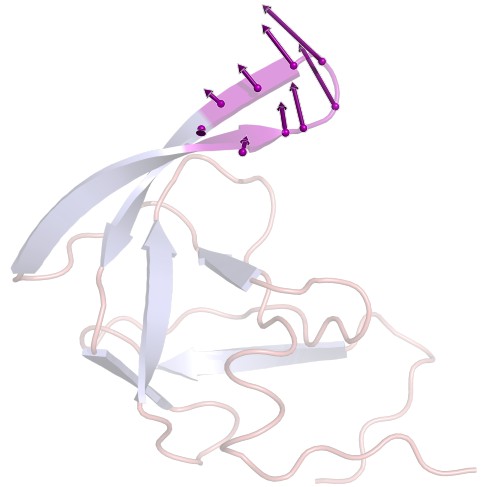

*Figure 11.* 1hhp target

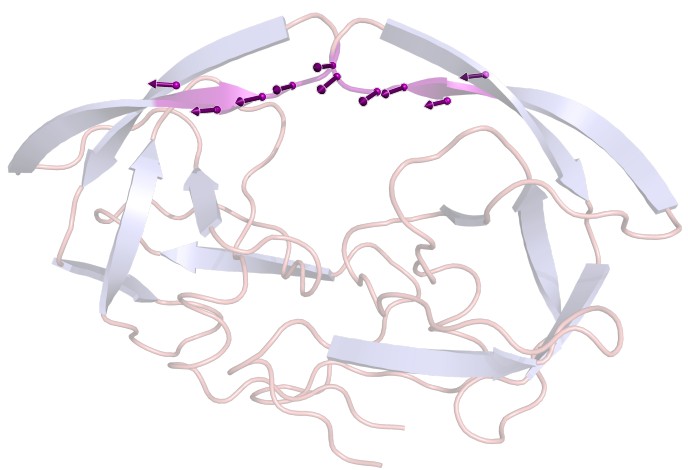

*Figure 12.* 1hhp assembly target

# E. Individual results for 3 sampling runs

### E.1. seed 1997

$\eta = 0.0$

| Target | 1hhp | | | 1exr | | | 1hhp_a | | | 4tgl | | |
|---|---|---|---|---|---|---|---|---|---|---|---|---|
| Dynamics-cond | tune | guid. | no | tune | guid. | no | tune | guid. | no | NMA-t | guid. | no |
| % designable | 60.9 | 63.6 | 65.5 | 53.6 | 66.4 | 85.5 | 20.9 | 46.4 | 50.0 | 10.0 | 16.4 | 16.4 |
| % w. sc-cossim > 0.9 | 11.8 | 4.5 | 0.9 | 10.0 | 3.6 | 0.9 | 0.0 | 0.9 | 0.0 | 0.0 | 0.0 | 0.0 |
| % w. sc-cossim > 0.8 | 20.9 | 15.5 | 8.2 | 30.9 | 18.2 | 5.5 | 4.5 | 3.6 | 0.0 | 0.0 | 0.0 | 0.0 |
| % w. sc-cossim > 0.7 | 27.3 | 19.1 | 12.7 | 40.0 | 27.3 | 13.6 | 6.4 | 4.5 | 0.0 | 1.8 | 0.9 | 0.0 |

$\eta = 0.1$

| Target | 1hhp | | | 1exr | | | 1hhp_a | | | 4tgl | | |
|---|---|---|---|---|---|---|---|---|---|---|---|---|
| Dynamics-cond | tune | guid. | no | tune | guid. | no | tune | guid. | no | NMA-t | guid. | no |
| % designable | 50.9 | 61.8 | 64.5 | 51.8 | 73.6 | 83.6 | 16.4 | 47.3 | 58.2 | 13.6 | 11.8 | 16.4 |
| % w. sc-cossim > 0.9 | 16.4 | 8.2 | 3.6 | 11.8 | 6.4 | 0.0 | 0.0 | 0.0 | 0.0 | 0.0 | 0.0 | 0.0 |
| % w. sc-cossim > 0.8 | 22.7 | 12.7 | 10.0 | 23.6 | 23.6 | 8.2 | 4.5 | 1.8 | 1.8 | 0.0 | 0.0 | 0.0 |
| % w. sc-cossim > 0.7 | 27.3 | 17.3 | 12.7 | 32.7 | 37.3 | 12.7 | 9.1 | 7.3 | 2.7 | 0.0 | 0.0 | 0.0 |

$\eta = 1.0$

| Target | 1hhp | | | 1exr | | | 1hhp_a | | | 4tgl | | |
|---|---|---|---|---|---|---|---|---|---|---|---|---|
| Dynamics-cond | tune | guid. | no | tune | guid. | no | tune | guid. | no | NMA-t | guid. | no |
| % designable | 41.8 | 57.3 | 60.0 | 53.6 | 58.2 | 69.1 | 10.0 | 36.4 | 36.4 | 10.0 | 15.5 | 13.6 |
| % w. sc-cossim > 0.9 | 5.5 | 0.9 | 0.0 | 8.2 | 0.0 | 0.0 | 0.0 | 0.0 | 0.0 | 0.0 | 0.0 | 0.0 |
| % w. sc-cossim > 0.8 | 12.7 | 12.7 | 3.6 | 20.9 | 6.4 | 0.9 | 0.9 | 0.9 | 0.0 | 0.0 | 0.0 | 0.0 |
| % w. sc-cossim > 0.7 | 19.1 | 17.3 | 11.8 | 30.0 | 19.1 | 1.8 | 4.5 | 2.7 | 2.7 | 0.0 | 0.9 | 1.8 |

**E.2. seed 1999**

$\eta = 0.0$

| Target | 1hhp | | | 1exr | | | 1hhp_a | | | 4tgl | | |
|---|---|---|---|---|---|---|---|---|---|---|---|---|
| Dynamics-cond | tune | guid. | no | tune | guid. | no | tune | guid. | no | tune | guid. | no |
| % designable | 55.5 | 69.1 | 73.6 | 55.5 | 68.2 | 80.9 | 16.4 | 45.5 | 52.7 | 16.4 | 10.0 | 13.6 |
| % w. sc-cossim $> 0.9$ | 13.6 | 9.1 | 1.8 | 10.9 | 10.0 | 1.8 | 0.9 | 0.9 | 0.0 | 0.0 | 0.0 | 0.0 |
| % w. sc-cossim $> 0.8$ | 21.8 | 16.4 | 7.3 | 24.5 | 23.6 | 7.3 | 5.5 | 0.9 | 0.9 | 2.7 | 0.0 | 0.0 |
| % w. sc-cossim $> 0.7$ | 36.4 | 24.5 | 9.1 | 31.8 | 29.1 | 12.7 | 8.2 | 3.6 | 1.8 | 5.5 | 1.8 | 0.0 |

$\eta = 0.1$

| Target | 1hhp | | | 1exr | | | 1hhp_a | | | 4tgl | | |
|---|---|---|---|---|---|---|---|---|---|---|---|---|
| Dynamics-cond | tune | guid. | no | tune | guid. | no | tune | guid. | no | tune | guid. | no |
| % designable | 56.4 | 60.9 | 77.3 | 57.3 | 70.0 | 82.7 | 19.1 | 46.4 | 48.2 | 11.8 | 8.2 | 10.9 |
| % w. sc-cossim $> 0.9$ | 8.2 | 6.4 | 4.5 | 9.1 | 3.6 | 0.9 | 2.7 | 0.9 | 0.0 | 0.0 | 0.0 | 0.0 |
| % w. sc-cossim $> 0.8$ | 21.8 | 12.7 | 10.9 | 21.8 | 18.2 | 9.1 | 3.6 | 2.7 | 2.7 | 0.0 | 0.0 | 0.0 |
| % w. sc-cossim $> 0.7$ | 28.2 | 19.1 | 18.2 | 32.7 | 27.3 | 13.6 | 7.3 | 6.4 | 3.6 | 1.8 | 0.0 | 0.0 |

$\eta = 1.0$

| Target | 1hhp | | | 1exr | | | 1hhp_a | | | 4tgl | | |
|---|---|---|---|---|---|---|---|---|---|---|---|---|
| Dynamics-cond | tune | guid. | no | tune | guid. | no | tune | guid. | no | tune | guid. | no |
| % designable | 44.5 | 56.4 | 56.4 | 47.3 | 60.0 | 69.1 | 19.1 | 39.1 | 42.7 | 7.3 | 11.8 | 13.6 |
| % w. sc-cossim $> 0.9$ | 6.4 | 3.6 | 1.8 | 4.5 | 0.9 | 0.0 | 0.9 | 0.0 | 0.9 | 0.0 | 0.0 | 0.0 |
| % w. sc-cossim $> 0.8$ | 16.4 | 9.1 | 5.5 | 19.1 | 10.9 | 1.8 | 4.5 | 1.8 | 0.9 | 0.9 | 0.0 | 0.0 |
| % w. sc-cossim $> 0.7$ | 26.4 | 14.5 | 10.0 | 29.1 | 19.1 | 9.1 | 6.4 | 4.5 | 3.6 | 1.8 | 0.9 | 0.0 |

**E.3. seed 2007**

$\eta = 0.0$

| Target | 1hhp | | | 1exr | | | 1hhp_a | | | 4tgl | | |
| Dynamics-cond | tune | guid. | no | tune | guid. | no | tune | guid. | no | tune | guid. | no |
|---|---|---|---|---|---|---|---|---|---|---|---|---|
| % designable | 58.2 | 69.1 | 79.1 | 50.9 | 71.8 | 79.1 | 23.6 | 50.0 | 52.7 | 8.2 | 16.4 | 9.1 |
| % w. sc-cossim > 0.9 | 11.8 | 9.1 | 0.9 | 7.3 | 5.5 | 0.9 | 2.7 | 0.0 | 0.0 | 0.0 | 0.0 | 0.0 |
| % w. sc-cossim > 0.8 | 18.2 | 14.5 | 6.4 | 22.7 | 20.9 | 1.8 | 10.9 | 1.8 | 0.9 | 0.0 | 0.0 | 0.0 |
| % w. sc-cossim > 0.7 | 23.6 | 24.5 | 13.6 | 27.3 | 33.6 | 8.2 | 12.7 | 7.3 | 1.8 | 0.0 | 1.8 | 0.9 |

$\eta = 0.1$

| Target | 1hhp | | | 1exr | | | 1hhp_a | | | 4tgl | | |
| Dynamics-cond | tune | guid. | no | tune | guid. | no | tune | guid. | no | tune | guid. | no |
|---|---|---|---|---|---|---|---|---|---|---|---|---|
| % designable | 53.6 | 66.4 | 69.1 | 52.7 | 59.1 | 77.3 | 18.2 | 52.7 | 57.3 | 7.3 | 17.3 | 14.5 |
| % w. sc-cossim > 0.9 | 8.2 | 5.5 | 0.9 | 9.1 | 7.3 | 0.9 | 1.8 | 0.0 | 0.0 | 0.0 | 0.0 | 0.0 |
| % w. sc-cossim > 0.8 | 16.4 | 14.5 | 9.1 | 21.8 | 22.7 | 3.6 | 4.5 | 0.0 | 0.0 | 0.0 | 0.9 | 0.0 |
| % w. sc-cossim > 0.7 | 24.5 | 17.3 | 14.5 | 30.9 | 28.2 | 9.1 | 8.2 | 2.7 | 0.9 | 0.0 | 0.9 | 0.9 |

$\eta = 1.0$

| Target | 1hhp | | | 1exr | | | 1hhp_a | | | 4tgl | | |
| Dynamics-cond | tune | guid. | no | tune | guid. | no | tune | guid. | no | tune | guid. | no |
|---|---|---|---|---|---|---|---|---|---|---|---|---|
| % designable | 41.8 | 61.8 | 55.5 | 38.2 | 60.9 | 66.4 | 18.2 | 34.5 | 33.6 | 7.3 | 14.5 | 12.7 |
| % w. sc-cossim > 0.9 | 6.4 | 4.5 | 0.9 | 1.8 | 1.8 | 0.9 | 0.0 | 0.0 | 0.0 | 0.0 | 0.0 | 0.0 |
| % w. sc-cossim > 0.8 | 12.7 | 12.7 | 9.1 | 11.8 | 11.8 | 2.7 | 1.8 | 0.0 | 0.0 | 0.0 | 0.9 | 0.0 |
| % w. sc-cossim > 0.7 | 17.3 | 16.4 | 10.9 | 18.2 | 20.9 | 8.2 | 6.4 | 0.9 | 1.8 | 1.8 | 0.9 | 0.0 |

