# OpenReview forum: "NMA-tune: Generating Highly Designable and Dynamics Aware Protein Backbones"
_ICML.cc/2025/Conference — ICML 2025 poster_

### Official Review · Reviewer_uR1t · 2025-03-13

**Overall Recommendation:** 3

**Summary:**

NMA-tune is a new method that incorporates dynamic information into protein design by conditioning backbone generation on the lowest normal mode of oscillation. It extends RFdiffusion as a plug-and-play component, improving the proportion of samples with high structural quality and desired dynamics. The approach addresses challenges in generating molecules with both designable quality and targeted motions. Molecular Dynamics simulations confirm the presence of the targeted modes.

**Claims And Evidence:**

In the introduction part, the author argued that NMA-guidance has relatively poor performance because of the balance between conditional and unconditional terms. However, it is hard to tell that the model gains performance because of trainable conditioning networks since additional loss terms are introduced in Section 4.2. Experimental evidence is needed to support this. Besides, even if they claim they could adjust the unconditional and conditional terms, they are still upscaling the conditional term in the experiments.

**Essential References Not Discussed:**

The authors have provided a fair related work discussion section, however, some of the statements are not correct. For example, NMA-guidance also has a trainable GVP based model for denoising. A short literature review of condition guided diffusion model papers is missing here too.

**Experimental Designs Or Analyses:**

The results presented in Section 4 are quite limited. Additionally, the section on evaluation metrics is poorly articulated; it only includes two metrics, which could be effectively conveyed by simply stating the relevant thresholds. In Section 5, the authors have altered the threshold for sc-RMSD for two targets, which compromises the fairness of the experiment. It would be more appropriate to select other, more suitable targets for experimentation.
Furthermore, the results in Table 1 indicate that the performance of “tune” is significantly inferior to that of “guid” on the designable metric. The authors need to demonstrate that this discrepancy is not due to an improperly tuned scaling factor affecting conditional and

**Methods And Evaluation Criteria:**

The novelty of the methodology lies in its ability to decouple the conditional score from the large diffusion model during training, eliminating the need to backpropagate through RFdiffusion. Additionally, the evaluation metrics currently only consider designability and sc-cossim; it would be important to also report on novelty and diversity. Moreover, the experiments are limited to only four targets, which may constrain the robustness of the findings presented in the paper.

**Other Comments Or Suggestions:**

Table 2 and Table 3 should have a head row to show which model is being used.

**Other Strengths And Weaknesses:**

Weaknesses:
The paper lacks organization, with an excessive amount of pages devoted to background and related work. It would be more effective to retain only the essential elements of this content in the main text.

**Questions For Authors:**

All questions are listed in the above sections.

**Relation To Broader Scientific Literature:**

This paper builds significantly upon the earlier work titled “Dynamics-informed Protein Design with Structure Conditioning,” presented at ICLR 2024. Many of the theoretical concepts are derived from that foundational research.

**Theoretical Claims:**

In Eq. 6 and Eq. 7, the author mentions that the Jacobian term can be disregarded. I am curious about how the model can accommodate this, considering that ( x_t ) is not input into the correction network. A more theoretical discussion on this topic would be appreciated.

---

> ### Author Rebuttal · Authors · 2025-03-31
>
> Thank you for carefully going through our manuscript and giving us your constructive feedback. As you mentioned, we tackle “challenges in generating molecules with both designable quality and targeted motions”, and we are happy to see you note the strength of the MD simulations evaluations. Let us address your questions one by one.
>
> 1. “[...] the results in Table 1 indicate that the performance of “tune” is significantly inferior to that of “guid” on the designable metric. The authors need to demonstrate that this discrepancy is not due to an improperly tuned scaling factor affecting conditional and …” (though the sentence seems unfinished, we guess it was supposed to say “conditional and unconditional terms”)
>
> It seems to us that a couple of your concerts across your review sections are strongly related to this final point you make. Please note that the final criterion for assessing the method’s performance is a simultaneous optimisation of both designability and NMA-loss. The guidance scale of NMA-guidance was fine-tuned for optimal performance in both of those metrics, and it struggled to achieve the balance between conditional and unconditional terms. Fine-tuning even the most advanced protein generative models is often a necessity (e.g. Proteina fine-tunes CFG and auto-guidance weights [1]). While NMA-tune also requires fine-tuning, it achieved much better balance in those metrics than NMA-guidance, which further motivates the usage of the trainable component.
>
> [1]. Proteina: Scaling Flow-based Protein Structure Generative Models,  https://arxiv.org/abs/2503.00710
>
> 2. “Additionally, the evaluation metrics currently only consider designability and sc-cossim; it would be important to also report on novelty and diversity.”
>
> Please take a look at our response to Reviewer FHJ7, where we provide those metrics.
>
> 3. “In Section 5, the authors have altered the threshold for sc-RMSD for two targets, which compromises the fairness of the experiment”
>
> Since NMA-tune and NMA-guidance are based on the same base model RFdiffusion, which struggles exactly in the same way to perform structure conditioning of the difficult targets, we believe the comparison of two dynamics-conditioning methods remains fair even with the adjusted sc-RMSD criterion. Ideally, we would evaluate using a standardised benchmark of dynamical-motifs, like RFdiffusion benchmark for static motifs. Currently, no such benchmark exists, and the number of targets we can use is limited.

---

> > ### Comment · Reviewer_uR1t · 2025-04-05
> >
> > Thank you for addressing my concerns. I have raised my scores.

---

> > > ### Author Response · Authors · 2025-04-08
> > >
> > > As the discussion period comes to an end, we would like to thank all the reviewers for the great effort they put into reviewing our work and for appreciating the scientific contributions of NMA-tune.

---

### Official Review · Reviewer_k2U2 · 2025-03-14

**Overall Recommendation:** 4

**Summary:**

The paper introduces NMA-tune, a plug-and-play modification to the RFDiffusion framework aimed at enhancing protein design by integrating Normal Mode Analysis (NMA)-inspired diffusion conditioning correction. The proposed method introduces a computationally efficient conditioning term that utilizes the fully denoised RFDiffusion sample prediction to correct the diffusion trajectory. The authors demonstrate an improvement in motif scaffolding tasks and analyze the approach’s performance from a Molecular Dynamics (MD) perspective.

**Claims And Evidence:**

The claims regarding improved motif scaffolding are supported by experiments involving three case-study proteins with well-documented hinge motions. However, the interpretation of the MD analysis results is not entirely clear, and additional references to tables and figures would strengthen the conclusions.

**Essential References Not Discussed:**

All essential references are discussed.

**Experimental Designs Or Analyses:**

The experimental section is strong. The authors present three case-study proteins with well-documented hinge motions in the literature. However, the results from the MD analysis could be better explained to highlight key findings. Furthermore, the paper would benefit from a direct comparison with other motif scaffolding approaches to establish a more comprehensive baseline.

**Methods And Evaluation Criteria:**

The proposed method aligns well with the problem statement. Still, a direct comparison with alternative motif scaffolding approaches, such as SMCDiff [a], would further substantiate the claim of performance improvement.

a. Diffusion probabilistic modeling of protein backbones in 3D for the motif-scaffolding problem, Trippe et al., 2022

**Other Comments Or Suggestions:**

I suggest to:
1. Include column descriptions in Tables 2 and 3, as they are not immediately obvious and require searching within the text.
2. Revise the MD analysis section to highlight key findings and provide references to tables for clarity.
3. Compare NMA-tune with other motif scaffolding approaches, such as SMCDiff, to strengthen the empirical evaluation.

**Other Strengths And Weaknesses:**

The paper is well-written but not always easy to follow. While the method description is clear, additional intuition and explanations for some experimental choices would improve clarity. Including more visualizations of generated proteins in the main text would enhance the presentation.

**Questions For Authors:**

All suggestions were stated in other sections.

**Relation To Broader Scientific Literature:**

The paper advances motif scaffolding and plausible protein structure generation by introducing a plug-and-play conditioning correction for RFDiffusion, guiding diffusion toward the lowest non-trivial normal mode. This novel and original approach opens a new direction for integrating physical constraints into generative models.

**Theoretical Claims:**

The mathematical description and algorithm appear correct.

---

> ### Author Rebuttal · Authors · 2025-03-31
>
> Thank you for providing your constructive critique of our work. We are happy to hear that you appreciate the strong points of our paper, particularly you note “the experimental section is strong. The authors present three case-study proteins with well-documented hinge motions in the literature.”  As you suggest, in the future we would like to expand NMA-tune to operate with other motif scaffolding approaches as well. While it is not feasible in this short rebuttal period, and the strength of our method lies in dynamics rather than just structure conditioning, using other motif scaffolding models would be a great contribution to the community.
>
> We will revise for clarity the Tables’ descriptions and the findings of the MD evaluation. Please let us know if you have any direct comments on the writing style of the results Section.

---

> > ### Comment · Reviewer_k2U2 · 2025-04-05
> >
> > I thank the authors for their response. I keep my current score.

---

> > > ### Author Response · Authors · 2025-04-08
> > >
> > > As the discussion period comes to an end, we would like to thank all the reviewers for the great effort they put into reviewing our work and for appreciating the scientific contributions of NMA-tune.

---

### Official Review · Reviewer_xx2G · 2025-03-14

**Overall Recommendation:** 1

**Summary:**

This paper introduces a training-based method to address the problem of dynamic-conditioned generation of proteins. Specifically, they replace the prior-guided term with a simpler, more computationally efficient one to improve sampling speed, and they introduce a small network to learn such conditioned mappings.

## update after rebuttal

I keep my original rating.

**Claims And Evidence:**

The authors claim that their tuning-based method offers superior performance and accelerates sampling. However, as shown in Table 1, the designability is notably worse compared to the guided method. Furthermore, there are no metrics provided that consider sampling speed. Given that the tuning-based method incurs additional training costs, I am not convinced that it is a better strategy than the guided approach.

**Essential References Not Discussed:**

None.

**Experimental Designs Or Analyses:**

I have examined the experiments and compared them with prior works:

1.	I am concerned about the performance gap between the tuning-based method and the prior guidance method. Although the paper asserts that the learned guidance term is better, the actual performance is worse.

2.	This paper evaluates methods on a very small subset of dynamic proteins, whereas prior work uses a much larger dataset (10,037 protein structures). I recommend evaluating the proposed method on this larger dataset.

3.	It would be valuable to include more methods for comparison. For instance, beyond RFDiffusion, other protein generative models such as Frameflow, Framediff, and Genie (as used in the NMA-Guidance paper) should be considered. Additionally, I suggest including other guidance methods, such as classifier-based, classifier-free, and loss-guidance methods.

4.	In the MD-evaluation experiment, there is a lack of baselines.

5.	I recommend incorporating more dynamic-related metrics in addition to protein design metrics, since the paper claims to achieve more accurate dynamic-conditioned generation.

**Methods And Evaluation Criteria:**

I question the necessity of training a network for this purpose. Moreover, the evaluation is conducted on a relatively small dataset, which may not accurately reflect real-world performance.

**Other Comments Or Suggestions:**

N/A

**Other Strengths And Weaknesses:**

1.	The writing and presentation need significant improvement. Sections 3 and 4 are particularly difficult to read, and there is no main figure in the method section.
2.	The intuition behind using a trainable network is unclear, and the results do not sufficiently support the claims.
3.	The experiments lack numerous baselines and should be conducted on a larger dataset.

**Questions For Authors:**

N/A

**Relation To Broader Scientific Literature:**

The idea of using loss-guidance is well-established (Song et al., ICML 2023), but this method employs a trainable network to approximate this term. The study of dynamic-conditioned protein generation has been explored in the NMA-guidance paper (Urszula et al., ICLR 2024).

**Theoretical Claims:**

I have reviewed the theoretical claims and find them to be correct.

---

> ### Author Rebuttal · Authors · 2025-03-31
>
> Thank you for providing your valuable feedback. Firstly, let us motivate again the usage of the trainable conditioner in NMA-tune. The analytical form of the NMA-loss that we use for loss-guidance might steer the generation into structures that have the ideal NMA-loss, but do not resemble proteins at all. Whether the balance between conditional and the unconditional terms can even be achieved under this loss formulation is the question we are tackling in this work. Most importantly, we show that NMA-tune achieves this balance, and optimises a number of metrics simultaneously, while NMA-guidance without a trainable conditioner struggles to do so.
>
> Next, let us address the other crucial points you make.
>
> 1. “However, as shown in Table 1, the designability is notably worse compared to the guided method.”
>
> The designability as measured by sc-RMSD is an important metric to understand the effects of conditioning. However, the most important metric that determines the method’s success is not just the number of designable samples, but the number of designable and successfully conditioned samples as measured by sc-RMSD and sc-cossim. **It is not enough to obtain a high quality sample - a sample must be both designable and meet the conditions,and with NMA-tune we show that we can achieve designability whilst meeting these conditions.**
>
> 2. “Furthermore, there are no metrics provided that consider sampling speed.”
>
> Please see our response to Reviewer FHJ7, where we provide the comparison of sampling speed, and show NMA-tune is much faster than NMA-guidance.
>
> “Moreover, the evaluation is conducted on a relatively small dataset, which may not accurately reflect real-world performance” and “This paper evaluates methods on a very small subset of dynamic proteins, whereas prior work uses a much larger dataset (10,037 protein structures).”
>
> **Please note that NMA-guidance does not evaluate the conditioning method on 10,037 structures.** Authors of NMA-guidance train their own unconditional protein generative model on 10,037 structures, while we use the already established and thoroughly evaluated RFdiffusion as our unconditional model. For the evaluation of the dynamics-conditioning on its own, the authors use 600 randomly selected targets, which roughly matches the number of samples we take for our carefully selected, real world targets. For that evaluation part, NMA-guidance also uses a different, less restrictive NMA-loss formulation (disregarding orientation of the conditioning residues).
>
> Regarding your concern that our method might not translate into real-world performance, please note that our MD evaluation is designed to tackle exactly this question. MD simulations are regarded as a very close proxy to the real-world behaviour of molecules, and to the best to our knowledge there are no better in silico approximations to the real-world motions than MD simulations. In our experiments Section, we show that the target motion is indeed present in the MD trajectory, therefore showing the success will be translated to real life.
>
> 3. “In the MD-evaluation experiment, there is a lack of baselines.”
>
> Our MD simulation experiments were designed to prove the link between presence of the target normal mode in the designed sample and the MD trajectory (close approximation to real life). Most importantly, they show that our *in silico* metrics (in which NMA-tune performs best) translate to real life, and therefore any designable, high quality and successfully dynamics-conditioned sample will exhibit the targeted motion in the MD trajectory. This is the crucial link for assessing  the method’s performance in biological tasks, but since 1) the link *in silico* metrics and MD trajectory is proven; 2) NMA-tune performs best as measured by *in silico* metrics; and 3) MD simulations are incredibly computationally expensive (one simulation can take about a day) and we have limited bandwidth, we deemed it sufficient to include only NMA-tune in the MD evaluation part.

---

### Official Review · Reviewer_FHJ7 · 2025-03-24

**Overall Recommendation:** 4

**Summary:**

The paper aims to propose a solution to conditioning protein structure generation on structural dynamics. The authors define protein structure dynamics as the lowest normal modes of oscillations computed with Normal Mode Analysis (NMA) and propose an efficient strategy to incorporate this information into existing generative models for protein structure. In particular, the authors demonstrate that pretrained, unconditional diffusion-based generative models, such as RFdiffusion, can be turned into dynamics-conditional ones via loss guidance without requiring retraining or fine-tuning, and call their framework NMA-tune.

The authors extend the work of Komorowska et al. (2024) and introduce an SO(3)-equivariant conditioner, which learns the conditional loss guidance term in contrast to the analytical approximation proposed previously in Komorowska et al. (2024). The conditioner essentially learns the set of translations that need to be applied to the unconditional noise in order to sample from the conditional probability distribution. The authors also propose a training strategy along with an SE(3)-invariant NMA-loss for their conditioner to solve the problem of generating structures with certain functional motifs that encode a movement of interest. In a series of experiments, the authors demonstrate the efficiency of their method and outperform a baseline.

## Update after rebuttal

I will keep my score. I believe this work makes an interesting contribution and consistently achieves better performance on relevant metrics.

**Claims And Evidence:**

Thoroughly conducted experiments convincingly demonstrate that the proposed
method outperforms a previously published baseline. The authors report informative
metrics on conditionally generated protein structures, such as secondary
structure composition, and assess the effect of conditioning on the well established
designability metric. They justify the selection of the proteins used
in experiments, conduct extensive Molecular Dynamics (MD) simulations to
cross-validate recapitulation of the NMA-derived lowest modes passed as a condition,
and clearly state their limitations.

**Essential References Not Discussed:**

Another recent work which focuses on high designability perhaps worth mentioning is
Wagner et al., Generating Highly Designable Proteins with Geometric Algebra Flow Matching, Neurips 2024.

**Experimental Designs Or Analyses:**

• Table 1: For reproducibility, I suggest adding full inference settings of
RFdiffusion beyond the noise scale (e.g., number of timesteps). It would
be informative to include novelty and diversity metrics (see example definitions
in [4]) for generated protein structures. Lower designability of
NMA-tune compared to unconditional generation might stem from novel
backbone geometries.

• Figures 2 and 3: Report Cα-Cα distances and secondary structure distributions
without filtering for designability. I would like to see distributions
for only designable samples.

• Inference Time: Absolute inference time for each sampling method
would be informative.

• Section 5.2: MolProbity and SwissModel scores may be unfamiliar to
some readers. An appendix explaining these metrics would be useful.

• Figure 1: Good visualization of the sc-cossim, but the colors of the arrows
should be adjusted for better readability, especially in Figure 1(b), where
it is difficult to see the direction of the arrows.

[4] Yim, Jason, et al. ”Fast protein backbone generation with se (3) flow matching.”
arXiv preprint arXiv:2310.05297 (2023).

**Methods And Evaluation Criteria:**

Yes. Although NMA, while being computationally efficient, only captures types
of harmonic motions about the equilibrium state, which might not be sufficient
to condition on biologically relevant, functional directional movements [1],
such as state-transitions performed during ligand binding [2] or catalysis [3].
The authors acknowledge this limitation and compare their results with MD
simulation-derived lowest modes in different time windows (Tables 2 and 3).
To the best of my knowledge, no precise and computationally efficient methods
exist to generate and incorporate reliable mid- to long-range protein structure
dynamics during training of the conditioner. Thus, the motivation to use NMA
is clear.

[1] Alexandrov, Vadim, et al. ”Normal modes for predicting protein motions: a
comprehensive database assessment and associated Web tool.” Protein science
14.3 (2005): 633-643.
[2] Deng, Hong, Nick Zhadin, and Robert Callender. ”Dynamics of protein ligand
binding on multiple time scales: NADH binding to lactate dehydrogenase.”
Biochemistry 40.13 (2001): 3767-3773.
[3] Schramm, Vern L. ”Enzymatic transition states and transition state analogues.”
Current opinion in structural biology 15.6 (2005): 604-613.

**Other Comments Or Suggestions:**

None

**Other Strengths And Weaknesses:**

Strengths:
• The paper is very well-written with smooth transitions between sections.
The authors state the limitations of the proposed method.

Weaknesses:
• The novelty of the theoretical contribution is limited.

**Questions For Authors:**

None

**Relation To Broader Scientific Literature:**

To the best of my knowledge, no existing generative models condition protein
structure generation on dynamics observables. The only relevant baseline is
NMA-guidance, which inspired the refinement proposed in this paper.

**Theoretical Claims:**

Although the authors mostly adopt theoretical foundations from previously published
work Komorowska et al. (2024), they propose to learn the conditional loss term directly from the
predictions of denoised protein structures made by the generative model via a
graph neural network. The authors demonstrate that this approach performs
better than estimating the guidance term analytically as in Komorowska et al. (2024).
The goal is to sample from the conditional posterior p(y | x0) , where y encodes
the eigenvectors of a Hessian matrix from the NMA computation. Instead of
learning the eigenvectors directly, the authors propose to learn a correction term
to the unconditional noise which best matches the eigenvectors of
the precomputed lowest modes.

---

> ### Author Rebuttal · Authors · 2025-03-31
>
> Thank you for reading our manuscript in great detail. We are glad to know that you find our work well-written and you appreciated the strength of our experimental evaluation.
>
> Thank you for pointing out that the “conducted experiments convincingly demonstrate that the proposed method outperforms a previously published baseline”. We are grateful for your feedback on how to improve the clarity of the manuscript. Please note that during the rebuttal period there is no option to upload a revised version of the paper, but we certainly will implement those changes in the future version of the manuscript. Nevertheless, we are happy to report the extra evaluations you would like to see here.
>
> 1. “Report Cα-Cα distances and secondary structure distributions without filtering for designability.”
>
> We recalculated the results presented in Figures 2 and 3 in the Appendix B, but using only designable samples. The number of designable samples is much lower than the total number of samples, therefore we recomputed results using designable samples from all 3 seeds (for the Figures 2 and 3 we used 110 samples). The  $C_{\alpha}$- $C_{\alpha}$ distances and secondary structure distributions are now as follows:
>
> For 1hhp assembly
>
> $C_{\alpha}$ dist (A): NMA-guid.:  3.7679 ± 0.0075, NMA-tune: 3.7616 ± 0.0080, Uncond.: 3.7679 ± 0.0075
>
> ### Secondary Structure Usage
>
> |        | NMA-guid.          | NMA-tune         | Uncond.          |
> |--------|--------------------|------------------|------------------|
> | Alpha  | 0.5338             | 0.4286           | 0.5532           |
> | Beta   | 0.2689             | 0.3378           | 0.2539           |
> | Coil   | 0.1974             | 0.2337           | 0.1929           |
>
> For 1exr
>
> $C_{\alpha}$ dist (A): NMA-guid.:  3.7767 ± 0.0032, NMA-tune: 3.7780 ± 0.0025, Uncond.: 3.7803 ± 0.0033
>
> ### Secondary Structure Usage
>
> |        | NMA-guid.          | NMA-tune         | Uncond.          |
> |--------|--------------------|------------------|------------------|
> | Alpha  | 0.8846             | 0.8783           | 0.8903           |
> | Beta   | 0.0027             | 0.0013           | 0.0015           |
> | Coil   | 0.1127             | 0.1204           | 0.1082           |
>
>
> Changes in the above statistic follow the same patterns as when calculated using all samples. $C_{\alpha}$ distances remain in the realistic range both for NMA-tune and NMA-guidance, and again both methods slightly disturb the secondary structure statistics, which is expected since conditioning induces a distribution shift.
>
> 2. “Absolute inference time for each sampling method would be informative.”
>
> We calculated the running time of the sampling loop (with the default RFdiffusion number of time steps equal to 50) for NMA-tune and NMA-guidance. Mean loop running time for each target on our Nvidia A100 80GB machine, averaged over 110 samples per target, was about 36 seconds for NMA-tune, and about 63 seconds for NMA-guidance, **which gives about 75% speedup**.
>
> 3. “It would be informative to include novelty and diversity metrics”
>
> According to your suggestion, we use Foldseek [1] and MaxCluster [2] to evaluate novelty and diversity for two targets: 1exr and 1hhp_assembly. For novelty,  we compute theTM-score to AFDB and PDB100 databases available at Foldseek server, and for each sample retain the max score. We then report the mean of those max TM-scores (the lower, the more novel samples).
>
> | Novelty (TM-scores)      | 1exr, $\eta$=0.0 | 1exr, $\eta$=1.0 | 1hhp_a, $\eta$=0.0 | 1hhp_a, $\eta$=1.0 |
> |--------------|----------------|----------------|---------------------|---------------------|
> | NMA-tune  | 0.68           | 0.64           | 0.52                | 0.53                |
> | NMA-guid. | 0.71           | 0.68           | 0.57                | 0.55                |
>
>
> While it seems that NMA-tune outperforms NMA-guidance by a narrow margin, novelty of both methods remains in the range comparable to other generative models (FrameDiff [3], FrameFlow [4]).
>
> We calculate diversity using MaxCluster with hierarchical clustering (single linkage method), in sequence independent mode, with a TM-score threshold 0.6. From the set of all 110 samples generated per target per noise scale for one seed, we take samples for $\eta=0.0$ and $\eta=1.0$ together, and discard the non-designable samples. Results for the remaining designable samples are as follows (clusters / num of designable samples):
>
> NMA-tune: 1exr target: 13/118; 1hhp_assembly target: 32/34
>
> NMA-guidance: 1exr target: 10/137; 1hhp_assembly target: 52/91
>
> As expected, diversity depends on the scaffolding target. Neither of the methods collapses to sample from a single cluster.
>
>
> [1] https://www.nature.com/articles/s41587-023-01773-0 \
> [2] https://www.sbg.bio.ic.ac.uk/maxcluster/ \
> [3] https://arxiv.org/pdf/2302.02277 \
> [4] https://arxiv.org/pdf/2302.02277

---

> > ### Comment · Reviewer_FHJ7 · 2025-04-04
> >
> > I thank the authors for their reponse.
> >
> > There seems to be a slight trend towards incorporating higher amount of coils in the scaffolds upon conditioning. It would be interesting to see a plot or a table with scRMSD for the whole scaffold and a motif for both NMA-tune ad NMA-guide compared with the average scRMSD of unconditional samples. This could help assessing whether sc-metrics are just borderline above the thresholds.
> >
> > I keep my positive rating.

---

> > > ### Author Response · Authors · 2025-04-08
> > >
> > > Thank you for pointing out that we should check whether the designable samples are not too close to the acceptance threshold. We performed a sanity check of designable samples for two targets at two $\eta$ values, and computed median sc-motif-RMSD and sc-RMSD. We observed that RMSD values are often safely lower than the acceptance threshold, and we confirmed that designable samples are not just 'barely' designable. In the tables we report median values in Angstrom.
> > > ### 1hhp  $\eta$=0.0
> > >
> > > | Method     | scRMSD | sc-motif-RMSD |
> > > |------------|---------------|-----------------------|
> > > | NMA-tune   | 0.864         | 0.684                 |
> > > | NMA-guid   | 0.706         | 0.645                 |
> > > | uncond     | 0.678         | 0.667                 |
> > >
> > > ### 1hhp  $\eta$=1.0
> > >
> > > | Method     | scRMSD | sc-motif-RMSD |
> > > |------------|---------------|-----------------------|
> > > | NMA-tune   | 0.935         | 0.688                 |
> > > | NMA-guid   | 0.816         | 0.737                 |
> > > | uncond     | 0.776         | 0.720                 |
> > >
> > > ### 1exr $\eta$=0.0
> > >
> > > | Method     | scRMSD | sc-motif-RMSD |
> > > |------------|---------------|-----------------------|
> > > | NMA-tune   | 0.714         | 0.406                 |
> > > | NMA-guid   | 0.732         | 0.411                 |
> > > | uncond     | 0.658         | 0.381                 |
> > >
> > > ### 1exr  $\eta$=1.0
> > >
> > > | Method     | scRMSD | sc-motif-RMSD |
> > > |------------|---------------|-----------------------|
> > > | NMA-tune   | 0.840         | 0.395                 |
> > > | NMA-guid   | 0.889         | 0.408                 |
> > > | uncond     | 0.771         | 0.342                 |
> > >
> > > Finally, we would like to thank you for the effort you put in reviewing our work, and for appreciating the scientific contributions of NMA-tune.

---

### Decision · Program_Chairs · 2025-05-01

**Decision:**

Accept (poster)

**Comment:**

The paper presents NMA-tune, a method for conditioning protein structure generation on structural dynamics. Reviewers had a mix of positive and critical views. Consensus points included the methodological soundness, strong experimental aspects, and good connection to the literature. Concerns include that reviewers are not convinced by the lower designability, questioned the need for a training network, and pointed out the small dataset size.  Overall, despite weaknesses like limited novelty and performance concerns, the paper's strengths, such as its experimental design and novel approach, are significant. The recommendation is to accept the paper with the condition that the authors address remaining issues in a revised version, including clarifying the MD analysis, evaluating on a larger dataset, and justifying the designability trade-off.